# Is Value Functions Estimation with Classification Plug-and-play for Offline Reinforcement Learning?

**Denis Tarasov**  *tarasovd@ethz.ch*
*Department of Computer Science*
*ETH Zürich*

**Kirill Brilliantov**  *kbrilliantov@ethz.ch*
*Department of Computer Science*
*ETH Zürich*

**Dmitrii Kharlapenko**  *dkharlapenko@ethz.ch*
*Department of Computer Science*
*ETH Zürich*

**Reviewed on OpenReview:** *https://openreview.net/forum?id=MHJlFCqXdA*

## Abstract

In deep Reinforcement Learning (RL), value functions are typically approximated using deep neural networks and trained via mean squared error regression objectives to fit the true value functions. Recent research has proposed an alternative approach, utilizing the cross-entropy classification objective, which has demonstrated improved performance and scalability of RL algorithms. However, existing study have not extensively benchmarked the effects of this replacement across various domains, as the primary objective was to demonstrate the efficacy of the concept across a broad spectrum of tasks, without delving into in-depth analysis. Our work seeks to empirically investigate the impact of such a replacement in an offline RL setup and analyze the effects of different aspects on performance. Through large-scale experiments conducted across a diverse range of tasks using different algorithms, we aim to gain deeper insights into the implications of this approach. Our results reveal that incorporating this change can lead to superior performance over state-of-the-art solutions for some algorithms in certain tasks, while maintaining comparable performance levels in other tasks, however for other algorithms this modification might lead to the dramatic performance drop. This findings are crucial for further application of classification approach in research and practical tasks[1].

## 1 Introduction

In the realm of deep Reinforcement Learning (RL), the conventional approach to approximating value functions has long relied on employing the Bellman optimality operator alongside mean squared error (MSE) regression objectives, owing to the continuous nature of the task at hand. However, insights from other domains of machine learning have illuminated the potential benefits of employing classification objectives even in scenarios where regression seems as a natural choice (Rothe et al., 2018; Rogez et al., 2019). This shift has been attributed to various hypotheses, including the stability of gradients (Imani et al., 2024), improved feature representation (Zhang et al., 2023), and implicit biases (Stewart et al., 2023).

---

[1]Our code is available at `https://github.com/DT6A/ClORL`

While the use of regression loss has yielded remarkable results in value-based RL (Silver et al., 2017), it also presents certain challenges and limitations (Kumar et al., 2020a; 2021; Agarwal et al., 2021; Lyle et al., 2022). Notably, recent research Farebrother et al. (2024) has demonstrated that replacing regression with classification for training value functions offers several advantages, including enhanced scalability, feature representation, performance, and robustness to noisy targets and non-stationarity. While this study broadly explores the effects of this replacement, it does not delve deeply into the ease of implementation or the impact of newly introduced hyperparameters.

Offline RL (Levine et al., 2020) represents a rapidly developing RL subfield, wherein the objective is to train an agent using a pre-collected dataset without direct interaction with the environment. In recent years, numerous algorithms have been developed to address this setup (Kumar et al., 2020b; Fujimoto & Gu, 2021; Kostrikov et al., 2021; An et al., 2021; Chen et al., 2021; Akimov et al., 2022; Yang et al., 2022; Ghasemipour et al., 2022; Nikulin et al., 2023; Tarasov et al., 2024a). The majority of these algorithms fall under the category of off-policy value-based approaches, owing to their alignment with the problem setup's requirements.

Given the prevalence of off-policy value-based approaches in offline RL, it becomes imperative to delve deeper into the potential impact of employing classification for value function training within this domain. While Farebrother et al. (2024) offer some experimental insights into offline RL tasks, we contend that there remain significant gaps to be addressed. Therefore, our study seeks to contribute to this area by conducting a thorough investigation into the utilization of classification objectives for value function training in offline RL scenarios.

This study aims to address the following questions through large-scale experiments conducted on a large range of tasks from the standard D4RL benchmark (Fu et al., 2020):

- Is classification a "plug-and-play replacement" for offline RL algorithms, and how does it impact performance?

- Does the use of classification objectives facilitate a more robust hyperparameter search?

- What is the impact of the hyperparameters introduced with classification?

- Does classification enable more efficient scaling of dense neural networks compared to regression?

## 2 Preliminaries

### 2.1 Offline Reinforcement Learning

The RL problem is conventionally framed as a Markov Decision Process characterized by a tuple $(S, A, P, R, \gamma)$, where: $S \subset \mathbb{R}^n$ denotes the state space, $A \subset \mathbb{R}^m$ represents the action space, $P : S \times A \to S$ is the transition function, $R : S \times A \to \mathbb{R}$ is the reward function, and $\gamma \in (0, 1)$ is the discount factor. The objective of RL is to find a policy $\pi : S \to A$ that maximizes the expected sum of discounted rewards: $\sum_{t=0}^{\infty} \gamma^t R(s_t, a_t)$. This entails the policy learning process, where the agent interacts with its environment by observing environmental states, taking actions in response, and receiving corresponding rewards.

Offline RL presents a departure from the traditional RL setup in that the agent relies solely on a pre-collected dataset $D$ collected by external agents. This paradigm introduces novel challenges, such as the estimation of value functions for out-of-distribution state-action pairs, thus giving rise to an entire subfield dedicated to addressing these challenges.

### 2.2 Q-function objective and classification

The Q-function stands as a pivotal concept in value-based RL, representing the expected return of a policy starting from state $s_t$ and taking action $a_t$: $Q(s_t, a_t) = \mathbb{E}_\pi[\sum_{k=0}^{\infty} \gamma^k R(s_{t+k}, a_{t+k})|s_t, a_t]$. Typically, value functions are trained using the Bellman optimality operator:

$$(\hat{\mathcal{T}}Q)(s, a, \hat{\theta}) = R(s, a) + \gamma \max_{a'} Q(s', a', \hat{\theta})$$

Here, $s'$ denotes the state observed after executing action $a$ at state $s$, and $\hat{\theta}$ (also known as target network) represents a copy with delayed updates of the parameter $\theta$ which parametrize the value function. During each training step, parameters $\theta$ are adjusted using temporal difference (TD) error with mean squared error (MSE):

$$\text{TD}_{\text{MSE}}(\theta) = \mathbb{E}_D((\hat{\mathcal{T}}Q)(s_t, a_t, \hat{\theta}) - Q(s_t, a_t, \theta))^2$$

We adopt the framework proposed by Farebrother et al. (2024) for replacing MSE with cross-entropy loss. The approach involves parameterizing the Q-function as a distribution over returns, segmented into $m$ bins with widths $\zeta$, where the first bin corresponds to the predefined $v_{min}$ value, and the last to $v_{max}$. The scalar value of the Q-function is then computed as:

$$Q(s, a, \theta) = \mathbb{E}[Z(s, a, \theta)], \quad Z(s, a, \theta) = \sum_{i=1}^{m} p_i(s, a, \theta)\delta_i,$$

where $p_i$ represents the probability of the $i$-th bin, and $\delta_i$ denotes the corresponding bin value. This formulation allows us to express the TD error using cross-entropy and utilize it for updating the value function parameters:

$$\text{TD}_{\text{CE}}(\theta) = \mathbb{E}_D \sum_{i=1}^{m} p_i^{\mathcal{T}}(s, a, \hat{\theta}) \log p_i(s, a, \theta)$$

Farebrother et al. (2024) demonstrated that the HL-Gauss method (Imani & White, 2018) is particularly effective for mapping continuous values into bins within this framework. HL-Gauss employs a normal distribution analog for mapping values into neighboring bins, with a hyperparameter $\sigma$ determining the distribution's breadth. The authors recommend tuning the ratio $\sigma/\zeta$ as a more interpretable hyperparameter, with a default value of 0.75 chosen based on statistical considerations and it has shown good empirical result.

## 3 Methodology

Our methodology is straightforward: we select several offline RL algorithms and adapt them for cross-entropy loss, as outlined in Section 2.2. By default, we determine the values of $v_{min}$ and $v_{max}$ by computing all possible discounted returns within a given dataset and identifying the minimum and maximum values, which aligns naturally with the offline RL setup. Herein, we provide brief descriptions of the algorithms employed in our study.

We selected three algorithms from the most prominent families of offline RL approaches: policy regularization, implicit regularization, and Q-function regularization. These categories encompass the majority of widely-recognized offline RL methods, as noted in prior work (Levine et al., 2020), making them representative choices for illustrating the general applicability of our framework. By selecting these specific algorithms, we aim to cover diverse approaches in offline RL while demonstrating that our classification loss can seamlessly integrate with various methods across these categories. The choice of particular algorithms is based on results from Tarasov et al. (2024b) where a wide range of offline RL algorithms were compared against each other. We provide pseudocodes for the algorithms with cross-entropy loss in Appendix B.

**Revisited BRAC (ReBRAC).** ReBRAC (Tarasov et al., 2024a) stands as a minimalist, state-of-the-art ensemble-free algorithm for both offline and offline-to-online RL. It employs MSE to penalize deviations from actions present in the dataset. Built upon TD3+BC (Fujimoto & Gu, 2021), ReBRAC incorporates several modifications that significantly enhance its performance. The Q-loss function takes the form:

$$\mathbb{E}_{(s,a,s',\hat{a}')\sim D,a'\sim \pi(s')}(Q(s,a,\theta) - [R(s,a) + \gamma(Q(s',a',\hat{\theta}) - \beta(a' - \hat{a}')^2)])^2$$

where $\beta$ denotes a penalty weight. Although only the target network part differs, the bin mapping is conducted in the same manner, with the penalty subtracted beforehand. We select ReBRAC as an exemplary algorithm with policy regularization (i.e. forcing the policy to commit actions simillar to dataset) due to its high performance, simplicity, and its encounter with the Q-function divergence problem while solving certain D4RL AntMaze tasks.

**Implicit Q-Learning (IQL).** IQL (Kostrikov et al., 2021) represents another competitive offline and offline-to-online RL algorithm. Its key advantage lies in its exclusion of out-of-distribution examples during training, a departure from most other offline RL approaches. This is achieved through the training of the V-function: $V(s_t) = \mathbb{E}_\pi[\sum_{k=0}^{\infty} \gamma^k R(s_{t+k}, a_{t+k})|s_t]$. The Q-function loss takes the form:

$$\mathbb{E}_{(s,a,s')\sim D}(Q(s,a,\theta) - [R(s,a) + \gamma V_\phi(s')])^2$$

Once again, the target component allows for classification without additional manipulations. Regularization is achieved through a specially designed V-function loss, without explicit penalties for the policy or value functions. We opt for IQL as it is the best example of algorithms within this family.

**Large-Batch SAC (LB-SAC).** LB-SAC (Nikulin et al., 2022) is an instance of the ensemble-based SAC-N algorithm (An et al., 2021), utilizing large-batch optimization to reduce ensemble size. SAC-N comprises $N$ Q-functions, each with the following loss:

$$\mathbb{E}_{(s,a,s')\sim D,a'\sim\pi(s')}(Q(s,a,\theta_i) - [R(s,a) + \gamma \min_{j=1..N} Q(s',a',\hat{\theta}_j)])^2$$

No additional adjustments are necessary to replace MSE with cross-entropy. The rationale behind this objective lies in the assumption that if Q-values follow a normal distribution $\mathcal{N}(\mu, \sigma)$ then:

$$\min_{j=1..N} Q(s,a,\hat{\theta}_j) \approx \mu(s,a) - \Phi^{-1}\left(\frac{N - \frac{\pi}{8}}{N - \frac{\pi}{4} + 1}\right)\sigma(s,a)$$

Ensemble-based approaches are known for their efficacy across many offline RL tasks. According to the above statement, SAC-N can be also considered as an example of offline RL algorithm with Q-function regularization. We opt for LB-SAC instead of the original SAC-N due to computational constraints that arise due to the huge ensemble sizes (up to 500 critics) required in original SAC-N.

## 4 Experimental Results

### 4.1 Experimental setup

We conducted our experiments using three sets of tasks from the D4RL benchmark (Fu et al., 2020): Gym-MuJoCo, AntMaze, and Adroit. We utilized all datasets within each set.

For ReBRAC in subsection 4.2, we employed the best hyperparameters as outlined in Tarasov et al. (2024a). When exploring parameter search in subsection 4.3, we utilized the same hyperparameter grids for ReBRAC and IQL as in Tarasov et al. (2024a) and used a custom grid for LB-SAC due to computational constraints. For a comprehensive overview of the experimental details and hyperparameters, refer to Appendix A and Appendix C.

The evaluation protocol is also taken from Tarasov et al. (2024a), where hyperparameters search is done using four random seeds and separate set of seeds is used for the evaluation with the exception that we utilized ten random seeds for the final evaluation for ReBRAC and IQL and only four seeds for LB-SAC, due to computational constraints. Note, that because of the chosen evaluation protocol tuned hyperparameters might perform worse than non-tuned which characterizes the sensitivity to the random initialization.

### 4.2 Is classification plug-and-play?

Our initial goal was to substitute MSE with cross-entropy without modifying any other aspects of the algorithms. We fixed the number of bins $m$ to 101, representing a reasonable number of classes. We set $\sigma/\zeta = 0.75$ by default as was proposed by Farebrother et al. (2024). Results for this modification are presented in the **CE** columns of Table 1 for Gym-MuJoCo, Table 2 for AntMaze, and Table 3 for Adroit. We also provide rliable (Agarwal et al., 2021) metrics which support all of the further claims in a more readable way in Appendix I. It is worth noting that, for ReBRAC in the AntMaze tasks here and further, we used

large-batch optimization, similar to that used in Gym-MuJoCo, which was previously hindered by Q-function divergence.

The original ReBRAC algorithm exhibited minimal performance variation with the introduction of cross-entropy in Gym-MuJoCo tasks, except for the random dataset where a notable performance drop was observed. However, on average, the score remained relatively stable. In the case of AntMaze, where ReBRAC faced Q-function divergence issues, the introduction of classification mitigated this problem, resulting in notable improvement in average performance. However, in Adroit, where ReBRAC struggled with overfitting due to small dataset sizes, classification did not alleviate this issue and led to decreased performance across most scenarios.

In contrast, both IQL and LB-SAC experienced a significant performance drop in Gym-MuJoCo tasks. IQL's performance in AntMaze dramatically decreased, rendering the algorithm unable to solve medium and large tasks with this modification. Meanwhile, LB-SAC's performance did not change significantly for AntMaze but improved for the umaze task. Notably, in the Adroit task, IQL's performance was significantly boosted in the pen environment, with relatively minor changes observed in other scenarios. However, LB-SAC's performance dropped across most tasks.

Our primary hypothesis for explaining the successful application of classification with ReBRAC and its failure with IQL and LB-SAC is that ReBRAC heavily relies on policy regularization. Consequently, plugging classification into Q networks does not strongly affect its offline RL component compared to the impact observed in IQL or LB-SAC.

Table 1: Average normalized score over the final evaluation and ten (four for LB-SAC) unseen random seeds on Gym-MuJoCo tasks. "hc" stands for "halfcheetah", "hp" stands for "hopper", "wl" stands for "walker2d", "r" stands for "random", "m" stands for "medium", "e" stands for "expert", "me" stands for "medium-expert", "mr" stands for "medium-replay", "fr" stands for "full-replay". $\pm$ denote stds across random seeds. **MSE** denotes original algorithm implementation, **CE** denotes the replacement of MSE with cross-entropy, **CE+AT** denotes cross-entropy with tuned algorithm parameters, **CE+CT** denotes cross-entropy with tuned classification parameters[2]. ReBRAC original scores are taken from Tarasov et al. (2024a).

| | ReBRAC | | | | IQL | | | | LB-SAC | | | |
|---|---|---|---|---|---|---|---|---|---|---|---|---|
| Task | MSE | CE | CE+AT | CE+CT | MSE | CE | CE+AT | CE+CT | MSE | CE | CE+AT | CE+CT |
| hc-r | 29.5 ± 1.5 | 13.4 ± 0.8 | 13.4 ± 0.8 | 13.0 ± 0.8 | 18.9 ± 1.0 | 1.9 ± 0.0 | 6.7 ± 0.9 | 3.9 ± 2.5 | 28.2 ± 1.4 | 10.0 ± 0.3 | 11.6 ± 0.2 | 11.1 ± 0.7 |
| hc-m | 65.6 ± 1.0 | 59.5 ± 0.7 | 63.8 ± 1.4 | 58.2 ± 11.2 | 49.5 ± 1.1 | 42.5 ± 0.3 | 42.6 ± 0.4 | 43.8 ± 0.3 | 64.5 ± 1.3 | 56.7 ± 2.1 | 63.6 ± 9.7 | 55.5 ± 2.7 |
| hc-e | 105.9 ± 1.7 | 103.2 ± 5.5 | 104.2 ± 2.5 | 103.7 ± 4.2 | 95.8 ± 2.2 | 92.9 ± 0.2 | 92.9 ± 0.4 | 93.5 ± 0.2 | 103.0 ± 1.5 | 103.9 ± 1.0 | 105.0 ± 1.3 | 104.5 ± 2.2 |
| hc-me | 101.1 ± 5.2 | 103.5 ± 4.3 | 103.2 ± 5.5 | 101.6 ± 6.4 | 92.3 ± 2.4 | 86.5 ± 4.0 | 83.3 ± 3.4 | 89.0 ± 3.8 | 104.5 ± 2.4 | 105.4 ± 2.0 | 105.2 ± 0.0 | 107.2 ± 1.3 |
| hc-mr | 51.0 ± 0.8 | 50.7 ± 0.7 | 53.1 ± 1.3 | 50.4 ± 2.3 | 45.2 ± 0.5 | 38.1 ± 1.8 | 39.9 ± 1.1 | 40.6 ± 1.8 | 52.8 ± 0.7 | 55.4 ± 0.9 | 54.4 ± 1.6 | 58.1 ± 1.3 |
| hc-fr | 82.1 ± 1.1 | 83.2 ± 1.5 | 82.4 ± 1.9 | 83.4 ± 0.9 | 75.5 ± 0.5 | 62.6 ± 1.1 | 64.5 ± 1.5 | 74.1 ± 0.6 | 79.0 ± 2.0 | 80.7 ± 0.3 | 81.3 ± 1.0 | 82.3 ± 1.1 |
| hp-r | 8.1 ± 2.4 | 8.1 ± 0.9 | 8.9 ± 2.1 | 9.1 ± 1.4 | 6.0 ± 2.8 | 14.2 ± 2.8 | 13.7 ± 3.3 | 13.8 ± 7.5 | 14.5 ± 11.5 | 8.2 ± 2.2 | 9.6 ± 0.4 | 10.9 ± 2.3 |
| hp-m | 102.0 ± 1.0 | 98.9 ± 9.4 | 101.7 ± 1.6 | 98.1 ± 7.5 | 54.8 ± 4.4 | 54.1 ± 2.0 | 52.9 ± 3.6 | 53.5 ± 1.8 | 90.0 ± 27.5 | 7.9 ± 0.8 | 10.5 ± 6.2 | 7.8 ± 1.0 |
| hp-e | 100.1 ± 8.3 | 107.9 ± 4.9 | 107.9 ± 4.9 | 110.8 ± 0.5 | 109.4 ± 1.8 | 110.5 ± 0.5 | 110.6 ± 0.3 | 110.2 ± 1.0 | 1.3 ± 0.0 | 21.6 ± 39.7 | 65.9 ± 51.9 | 12.3 ± 13.1 |
| hp-me | 107.0 ± 6.4 | 111.6 ± 0.5 | 111.2 ± 0.3 | 111.6 ± 0.5 | 88.5 ± 16.5 | 64.0 ± 10.2 | 69.3 ± 23.9 | 102.0 ± 8.6 | 111.3 ± 0.3 | 14.9 ± 7.0 | 12.5 ± 7.2 | 54.7 ± 37.6 |
| hp-mr | 98.1 ± 5.3 | 98.7 ± 3.0 | 94.1 ± 12.5 | 98.7 ± 3.0 | 95.9 ± 6.4 | 71.5 ± 10.1 | 27.2 ± 4.5 | 67.2 ± 8.8 | 63.0 ± 48.1 | 66.9 ± 42.8 | 100.6 ± 5.5 | 87.4 ± 29.6 |
| hp-fr | 107.1 ± 0.4 | 108.2 ± 0.3 | 108.3 ± 0.2 | 108.3 ± 0.5 | 107.2 ± 0.6 | 41.9 ± 4.8 | 53.9 ± 9.5 | 103.9 ± 0.6 | 107.0 ± 0.7 | 65.6 ± 50.2 | 109.9 ± 0.4 | 109.1 ± 1.8 |
| wl-r | 18.4 ± 4.5 | 8.8 ± 4.4 | 12.1 ± 9.1 | 7.2 ± 5.1 | 6.4 ± 6.3 | 3.9 ± 2.1 | 9.3 ± 7.6 | 6.3 ± 8.1 | 21.7 ± 0.0 | 20.1 ± 2.9 | 21.8 ± 0.0 | 21.5 ± 0.2 |
| wl-m | 82.5 ± 3.6 | 85.1 ± 2.7 | 85.1 ± 2.7 | 85.2 ± 1.0 | 83.2 ± 1.2 | 79.9 ± 1.3 | 81.3 ± 1.7 | 80.9 ± 1.5 | 89.3 ± 5.3 | 89.6 ± 10.7 | 93.7 ± 7.3 | 98.0 ± 1.3 |
| wl-e | 112.3 ± 0.2 | 112.7 ± 0.1 | 112.7 ± 0.1 | 112.7 ± 0.1 | 113.8 ± 0.2 | 108.5 ± 0.2 | 108.5 ± 0.2 | 109.5 ± 0.2 | 114.2 ± 0.4 | 59.8 ± 45.0 | 107.6 ± 1.2 | 112.5 ± 0.2 |
| wl-me | 111.6 ± 0.3 | 111.7 ± 0.1 | 111.9 ± 0.2 | 112.0 ± 0.3 | 112.3 ± 0.6 | 92.2 ± 6.5 | 108.6 ± 1.0 | 109.4 ± 0.1 | 110.6 ± 0.4 | 73.1 ± 18.1 | 94.0 ± 33.2 | 109.6 ± 8.8 |
| wl-mr | 77.3 ± 7.9 | 85.2 ± 6.7 | 84.4 ± 5.9 | 84.0 ± 5.0 | 82.0 ± 7.9 | 59.1 ± 8.8 | 67.2 ± 5.1 | 82.6 ± 3.8 | 92.6 ± 2.7 | 90.9 ± 5.3 | 95.3 ± 6.0 | 99.8 ± 1.9 |
| wl-fr | 102.2 ± 1.7 | 101.8 ± 5.6 | 108.7 ± 5.0 | 102.4 ± 2.9 | 97.7 ± 1.4 | 85.3 ± 3.3 | 83.9 ± 1.9 | 93.6 ± 1.0 | 102.1 ± 1.0 | 110.4 ± 1.9 | 109.4 ± 1.3 | 110.5 ± 2.4 |
| Avg | 81.2 | 80.6 | 81.5 | 80.6 | 74.1 | 61.6 | 62.0 | 70.9 | 74.9 | 57.8 | 69.5 | 69.6 |

## 4.3 Algorithms hyperparameters search with classification

The subsequent step in our investigation involved determining whether better performance could be achieved by tuning the hyperparameters of the original algorithms. The evaluation of the best hyperparameter sets can be found under the **CE+AT** columns in Table 1, Table 2, and Table 3 (see also Appendix I).

In Gym-MuJoCo tasks, these hyperparameter adjustments yielded slight improvements in ReBRAC's average performance, although random datasets remained problematic. Similarly, in Adroit tasks, performance saw

---

[2]These notations are reused further.

Table 2: Average normalized score over the final evaluation and ten (four for LB-SAC) unseen random seeds on AntMaze tasks. "um" stands for "umaze", "med" stands for "medium", "lrg" stands for "large", "p" stands for "play", "d" stands for "diverse". ± denote stds across random seeds. ReBRAC original scores are taken from Tarasov et al. (2024a).

| | ReBRAC | | | | IQL | | | | LB-SAC | | | |
|---|---|---|---|---|---|---|---|---|---|---|---|---|
| Task | MSE | CE | CE+AT | CE+CT | MSE | CE | CE+AT | CE+CT | MSE | CE | CE+AT | CE+CT |
| um | $97.8 \pm 1.0$ | $98.0 \pm 2.1$ | $98.0 \pm 1.6$ | $98.8 \pm 1.1$ | $79.1 \pm 6.8$ | $50.0 \pm 4.6$ | $49.6 \pm 8.1$ | $53.0 \pm 6.2$ | $18.25 \pm 35.8$ | $41.0 \pm 33.2$ | $36.0 \pm 33.4$ | $57.5 \pm 12.7$ |
| um-d | $88.3 \pm 13.0$ | $93.0 \pm 4.5$ | $91.5 \pm 8.0$ | $94.5 \pm 4.1$ | $72.5 \pm 4.5$ | $48.7 \pm 6.9$ | $47.4 \pm 5.3$ | $48.8 \pm 5.3$ | $0.0 \pm 0.0$ | $0.0 \pm 0.0$ | $0.2 \pm 0.5$ | $0.0 \pm 0.0$ |
| med-p | $84.0 \pm 4.2$ | $88.0 \pm 6.3$ | $90.5 \pm 3.8$ | $88.1 \pm 5.1$ | $73.8 \pm 5.4$ | $0.2 \pm 0.4$ | $0.1 \pm 0.3$ | $0.4 \pm 0.6$ | $0.0 \pm 0.0$ | $0.0 \pm 0.0$ | $0.0 \pm 0.0$ | $0.0 \pm 0.0$ |
| med-d | $76.3 \pm 13.5$ | $84.8 \pm 9.3$ | $76.6 \pm 14.8$ | $90.0 \pm 4.8$ | $74.8 \pm 3.8$ | $0.4 \pm 0.9$ | $0.3 \pm 0.6$ | $0.3 \pm 0.4$ | $0.0 \pm 0.0$ | $0.0 \pm 0.0$ | $0.0 \pm 0.0$ | $0.0 \pm 0.0$ |
| lrg-p | $60.4 \pm 26.1$ | $85.9 \pm 6.3$ | $87.0 \pm 4.3$ | $87.8 \pm 3.4$ | $41.3 \pm 7.0$ | $0.0 \pm 0.0$ | $0.0 \pm 0.0$ | $0.0 \pm 0.0$ | $0.0 \pm 0.0$ | $0.0 \pm 0.0$ | $0.0 \pm 0.0$ | $0.0 \pm 0.0$ |
| lrg-d | $54.4 \pm 25.1$ | $87.1 \pm 4.1$ | $86.6 \pm 4.6$ | $81.9 \pm 8.2$ | $23.7 \pm 5.6$ | $0.0 \pm 0.0$ | $0.0 \pm 0.0$ | $0.0 \pm 0.0$ | $0.0 \pm 0.0$ | $0.0 \pm 0.0$ | $0.0 \pm 0.0$ | $0.0 \pm 0.0$ |
| Avg | 76.8 | 89.4 | 88.3 | 90.1 | 60.8 | 16.5 | 16.2 | 17.0 | 3.0 | 6.8 | 6.0 | 9.5 |

Table 3: Average normalized score over the final evaluation and ten (four for LB-SAC) unseen random seeds on Adroit tasks. "ham" stands for "hammer", "rel" stands for "relocate", "h" stands for "human", "c" stands for "cloned", "e" stands for "expert". ± denote stds across random seeds. ReBRAC original scores are taken from Tarasov et al. (2024a).

| | ReBRAC | | | | IQL | | | | LB-SAC | | | |
|---|---|---|---|---|---|---|---|---|---|---|---|---|
| Task | MSE | CE | CE+AT | CE+CT | MSE | CE | CE+AT | CE+CT | MSE | CE | CE+AT | CE+CT |
| pen-h | $103.5 \pm 14.1$ | $102.3 \pm 10.2$ | $95.8 \pm 15.5$ | $97.5 \pm 13.3$ | $21.1 \pm 16.8$ | $108.2 \pm 8.4$ | $107.2 \pm 13.0$ | $93.6 \pm 8.7$ | $4.5 \pm 2.6$ | $7.1 \pm 5.6$ | $20.1 \pm 19.6$ | $5.8 \pm 8.8$ |
| pen-c | $91.8 \pm 21.7$ | $90.3 \pm 14.2$ | $94.0 \pm 18.3$ | $100.6 \pm 16.2$ | $13.2 \pm 21.6$ | $12.5 \pm 14.7$ | $99.9 \pm 21.7$ | $11.1 \pm 15.0$ | $26.1 \pm 5.3$ | $20.0 \pm 4.5$ | $20.8 \pm 5.0$ | $22.0 \pm 8.0$ |
| pen-e | $154.1 \pm 5.4$ | $155.0 \pm 5.8$ | $152.9 \pm 4.4$ | $155.9 \pm 4.8$ | $60.2 \pm 37.7$ | $134.9 \pm 7.0$ | $141.1 \pm 8.5$ | $141.3 \pm 4.3$ | $130.5 \pm 16.8$ | $38.0 \pm 13.2$ | $62.0 \pm 9.3$ | $43.6 \pm 29.8$ |
| door-h | $0.0 \pm 0.0$ | $0.0 \pm 0.0$ | $0.0 \pm 0.0$ | $0.0 \pm 0.1$ | $5.9 \pm 2.5$ | $4.3 \pm 1.0$ | $4.6 \pm 2.9$ | $3.4 \pm 2.1$ | $-0.2 \pm 0.1$ | $-0.2 \pm 0.1$ | $-0.2 \pm 0.1$ | $-0.2 \pm 0.1$ |
| door-c | $1.1 \pm 2.6$ | $0.0 \pm 0.0$ | $0.2 \pm 0.4$ | $0.0 \pm 0.0$ | $0.2 \pm 0.3$ | $0.3 \pm 0.3$ | $2.4 \pm 1.1$ | $1.2 \pm 0.7$ | $0.0 \pm 0.0$ | $0.2 \pm 0.5$ | $0.5 \pm 1.0$ | $0.0 \pm 0.0$ |
| door-e | $104.6 \pm 2.4$ | $105.7 \pm 1.5$ | $103.4 \pm 4.2$ | $105.0 \pm 2.4$ | $105.4 \pm 2.0$ | $106.1 \pm 0.5$ | $103.0 \pm 3.0$ | $105.7 \pm 1.6$ | $95.0 \pm 8.6$ | $70.6 \pm 33.8$ | $76.5 \pm 5.5$ | $68.7 \pm 19.4$ |
| ham-h | $0.2 \pm 0.2$ | $0.1 \pm 0.1$ | $0.4 \pm 0.3$ | $0.1 \pm 0.1$ | $1.7 \pm 0.8$ | $1.5 \pm 0.7$ | $2.1 \pm 1.9$ | $2.4 \pm 1.6$ | $0.1 \pm 0.0$ | $0.0 \pm 0.0$ | $0.1 \pm 0.0$ | $0.1 \pm 0.0$ |
| ham-c | $6.7 \pm 3.7$ | $9.7 \pm 10.4$ | $3.9 \pm 4.9$ | $6.1 \pm 8.8$ | $0.2 \pm 0.0$ | $1.0 \pm 0.7$ | $1.3 \pm 0.6$ | $2.6 \pm 4.4$ | $20.2 \pm 16.8$ | $13.6 \pm 15.4$ | $0.0 \pm 0.0$ | $19.9 \pm 21.7$ |
| ham-e | $133.8 \pm 0.7$ | $122.9 \pm 19.7$ | $120.1 \pm 24.9$ | $134.2 \pm 1.0$ | $129.5 \pm 0.2$ | $129.5 \pm 0.1$ | $129.5 \pm 0.1$ | $130.5 \pm 0.2$ | $76.6 \pm 59.5$ | $91.1 \pm 10.5$ | $96.0 \pm 9.9$ | $89.2 \pm 11.2$ |
| rel-h | $0.0 \pm 0.0$ | $0.0 \pm 0.0$ | $0.0 \pm 0.0$ | $0.0 \pm 0.0$ | $0.1 \pm 0.1$ | $0.1 \pm 0.0$ | $0.1 \pm 0.1$ | $0.1 \pm 0.0$ | $0.0 \pm 0.0$ | $-0.1 \pm 0.0$ | $0.0 \pm 0.0$ | $-0.1 \pm 0.0$ |
| rel-c | $0.9 \pm 1.6$ | $0.4 \pm 0.5$ | $0.3 \pm 0.3$ | $0.2 \pm 0.2$ | $0.1 \pm 0.1$ | $0.1 \pm 0.1$ | $0.1 \pm 0.1$ | $0.1 \pm 0.0$ | $0.0 \pm 0.0$ | $-0.1 \pm 0.0$ | $-0.1 \pm 0.0$ | $-0.1 \pm 0.0$ |
| rel-e | $106.6 \pm 3.2$ | $107.8 \pm 3.3$ | $107.2 \pm 2.8$ | $108.4 \pm 2.5$ | $108.2 \pm 0.9$ | $105.7 \pm 1.8$ | $109.4 \pm 0.7$ | $106.5 \pm 1.8$ | $26.7 \pm 18.8$ | $5.3 \pm 3.5$ | $5.1 \pm 3.5$ | $2.5 \pm 4.3$ |
| Avg w/o e | 25.5 | 25.3 | 24.3 | 25.5 | 5.3 | 16.0 | 27.2 | 14.3 | 6.3 | 5.0 | 5.1 | 5.9 |
| Avg | 58.6 | 57.8 | 56.5 | 59.0 | 37.1 | 50.3 | 58.3 | 42.5 | 31.6 | 20.4 | 23.4 | 20.9 |

some improvement but still fell short of the original algorithm's performance. Notably, for AntMaze tasks, the hyperparameter adjustments did not improve performance in this domain when compared to just plugging cross-entropy.

For IQL and LB-SAC, tuning the algorithms' hyperparameters with a cross-entropy objective helped alleviate underperformance in Gym-MuJoCo tasks, although random datasets remained a common issue. However, this approach did not yield significant improvements for AntMaze tasks, and only marginally improved average performance in Adroit.

Following the methodology outlined by Kurenkov & Kolesnikov (2022), we investigated whether algorithms utilizing classification offered superior hyperparameter search under uniform policy selection on D4RL tasks, using Expected Online Performance (EOP). The results are presented in Table 4 under the **CE+AT** rows. It is evident that for ReBRAC, classification facilitated much better hyperparameter search in the AntMaze domain, marginally improved search for Gym-MuJoCo, and showed slight degradation for Adroit tasks. IQL benefited only in the Adroit domain, while LB-SAC showed a slight improvement in AntMaze tasks.

## 4.4 What is the impact of classification parameters?

In Farebrother et al. (2024), the authors conducted a limited study on the influence of specific classification hyperparameters, only examining if $\sigma/\zeta = 0.75$ was suitable across varying numbers of bins $m$, using online RL tasks. Drawing inspiration from their work, we selected a set of $m$ values: $21, 51, 101, 201, 401$, and a set of $\sigma/\zeta$ values: $0.55, 0.65, 0.75, 0.85$. Subsequently, we conducted experiments using all possible pairs of these parameters, while maintaining the parameters of the original algorithms.

Evaluation results, presented in Table 1, Table 2, and Table 3 under the **CE+CT** columns (see also Appendix I), showcased notable differences in the impact of tuning classification parameters across algorithms. Specifically, ReBRAC exhibited improved performance when classification hyperparameters were fine-tuned, surpassing the original version's efficacy and achieving new state-of-the-art performance on AntMaze. In contrast, IQL and LB-SAC did not consistently benefit from classification parameter tuning compared to algorithm-specific adjustments.

EOP for this hyperparameter search is also provided in Table 4 under the **+CE+CT** rows. It's evident that when having good hyperparameters for original algorithm tuning classification parameters yields greater benefits than tuning algorithm-specific hyperparameters, as expected, with the only exception of IQL on Adroit tasks. Additionally, this search converges quickly, with training approximately three different policies often sufficing to achieve near-optimal performance, a highly advantageous property for real-world application of offline RL.

Table 4: Expected Online Performance (Kurenkov & Kolesnikov, 2022) under uniform policy selection aggregated over D4RL domains across four training seeds. This demonstrates the sensitivity to the choice of hyperparameters given a certain budget for online evaluation. **+CE+AT** denotes cross-entropy with tuned algorithm parameters, **+CE+CT** denotes cross-entropy with tuned classification parameters, **+CE+MT** denotes cross-entropy with mixed parameters tuning.

| Domain | Algorithm | 1 policy | 2 policies | 3 policies | 5 policies | 10 policies [3] | 15 policies | 18 policies | 20 policies |
|---|---|---|---|---|---|---|---|---|---|
| Gym-MuJoCo | **ReBRAC** | $62.0 \pm 17.1$ | $70.6 \pm 9.9$ | $73.3 \pm 5.5$ | $74.8 \pm 2.1$ | $75.6 \pm 0.8$ | $75.8 \pm 0.6$ | $75.9 \pm 0.6$ | $76.0 \pm 0.5$ |
| | **ReBRAC+CE+AT** | $64.1 \pm 15.4$ | $71.7 \pm 8.7$ | $74.0 \pm 4.8$ | $75.4 \pm 1.9$ | $76.2 \pm 1.1$ | $76.6 \pm 1.0$ | $76.7 \pm 0.9$ | $76.8 \pm 0.8$ |
| | **ReBRAC+CE+CT** | $78.4 \pm 1.4$ | $79.2 \pm 1.2$ | $79.7 \pm 1.0$ | $80.1 \pm 0.7$ | $80.5 \pm 0.5$ | $80.6 \pm 0.4$ | $80.7 \pm 0.4$ | $80.7 \pm 0.3$ |
| | **ReBRAC+CE+MT** | $62.0 \pm 15.5$ | $70.1 \pm 10.5$ | $73.3 \pm 6.6$ | $75.3 \pm 2.9$ | $76.4 \pm 0.9$ | $76.6 \pm 0.5$ | $76.7 \pm 0.3$ | - |
| | **IQL** | $62.3 \pm 9.8$ | $67.6 \pm 6.0$ | $69.5 \pm 3.8$ | $70.9 \pm 1.9$ | $71.6 \pm 0.7$ | $71.8 \pm 0.4$ | $71.9 \pm 0.3$ | $71.9 \pm 0.3$ |
| | **IQL+CE+AT** | $55.7 \pm 4.2$ | $58.1 \pm 3.6$ | $59.3 \pm 2.8$ | $60.5 \pm 1.8$ | $61.4 \pm 0.9$ | $61.7 \pm 0.7$ | $61.8 \pm 0.6$ | $61.8 \pm 0.6$ |
| | **IQL+CE+CT** | $58.4 \pm 7.2$ | $62.5 \pm 5.6$ | $64.4 \pm 4.3$ | $66.1 \pm 2.6$ | $67.2 \pm 1.3$ | $67.6 \pm 1.0$ | $67.8 \pm 0.9$ | $67.9 \pm 0.9$ |
| | **IQL+CE+MT** | $65.2 \pm 3.6$ | $67.2 \pm 2.7$ | $68.1 \pm 2.0$ | $68.9 \pm 1.2$ | $69.5 \pm 0.6$ | $69.7 \pm 0.4$ | $69.7 \pm 0.3$ | - |
| | **LB-SAC** | $52.5 \pm 13.8$ | $59.5 \pm 8.3$ | $62.0 \pm 5.5$ | $64.1 \pm 3.8$ | - | - | - | - |
| | **LB-SAC+CE+AT** | $48.9 \pm 11.3$ | $54.4 \pm 7.0$ | $56.4 \pm 5.2$ | $58.3 \pm 4.1$ | - | - | - | - |
| | **LB-SAC+CE+CT** | $63.4 \pm 3.7$ | $65.4 \pm 2.4$ | $66.2 \pm 1.7$ | $66.9 \pm 1.0$ | $67.3 \pm 0.6$ | - | - | - |
| | **LB-SAC+CE+MT** | $44.4 \pm 11.1$ | $50.5 \pm 7.9$ | $53.0 \pm 5.4$ | $55.0 \pm 2.9$ | $56.3 \pm 1.5$ | $56.8 \pm 1.2$ | $57.0 \pm 1.0$ | - |
| AntMaze | **ReBRAC** | $67.9 \pm 10.0$ | $73.6 \pm 7.4$ | $76.1 \pm 5.5$ | $78.3 \pm 3.4$ | $79.9 \pm 1.7$ | $80.4 \pm 1.1$ | - | - |
| | **ReBRAC+CE+AT** | $84.8 \pm 4.9$ | $87.5 \pm 3.5$ | $88.7 \pm 2.5$ | $89.7 \pm 1.5$ | $90.4 \pm 0.8$ | $90.7 \pm 0.5$ | - | - |
| | **ReBRAC+CE+CT** | $83.8 \pm 10.3$ | $88.3 \pm 4.7$ | $89.4 \pm 2.2$ | $90.0 \pm 0.9$ | $90.4 \pm 0.6$ | $90.6 \pm 0.4$ | $90.7 \pm 0.4$ | $90.7 \pm 0.3$ |
| | **ReBRAC+CE+MT** | $63.5 \pm 33.0$ | $79.8 \pm 22.3$ | $85.8 \pm 13.7$ | $89.2 \pm 5.2$ | $90.8 \pm 1.4$ | $91.2 \pm 0.9$ | $91.3 \pm 0.7$ | - |
| | **IQL** | $21.1 \pm 5.4$ | $24.1 \pm 4.7$ | $25.7 \pm 4.2$ | $27.5 \pm 3.6$ | $29.5 \pm 2.7$ | $30.4 \pm 2.0$ | $30.7 \pm 1.7$ | $30.9 \pm 1.5$ |
| | **IQL+CE+AT** | $15.6 \pm 0.9$ | $16.1 \pm 0.8$ | $16.4 \pm 0.7$ | $16.7 \pm 0.7$ | $17.0 \pm 0.6$ | $17.2 \pm 0.5$ | $17.3 \pm 0.4$ | $17.3 \pm 0.4$ |
| | **IQL+CE+CT** | $16.2 \pm 0.7$ | $16.6 \pm 0.7$ | $16.8 \pm 0.7$ | $17.1 \pm 0.7$ | $17.5 \pm 0.6$ | $17.7 \pm 0.6$ | $17.8 \pm 0.5$ | $17.8 \pm 0.5$ |
| | **IQL+CE+MT** | $23.3 \pm 9.3$ | $28.1 \pm 9.9$ | $31.2 \pm 9.6$ | $35.1 \pm 8.3$ | $39.2 \pm 5.1$ | $40.6 \pm 3.0$ | $41.0 \pm 2.3$ | - |
| | **LB-SAC** | $1.0 \pm 1.4$ | $1.6 \pm 1.6$ | $2.1 \pm 1.6$ | $2.8 \pm 1.4$ | - | - | - | - |
| | **LB-SAC+CE+AT** | $3.2 \pm 3.9$ | $5.1 \pm 3.9$ | $6.3 \pm 3.4$ | $7.6 \pm 2.2$ | - | - | - | - |
| | **LB-SAC+CE+CT** | $7.9 \pm 1.5$ | $8.7 \pm 1.1$ | $9.1 \pm 0.9$ | $9.4 \pm 0.5$ | $9.6 \pm 0.3$ | - | - | - |
| | **LB-SAC+CE+MT** | $2.8 \pm 3.7$ | $4.6 \pm 4.0$ | $6.0 \pm 3.8$ | $7.6 \pm 3.1$ | $9.2 \pm 1.7$ | $9.7 \pm 1.0$ | $9.8 \pm 0.8$ | - |
| Adroit | **ReBRAC** | $44.1 \pm 18.4$ | $53.2 \pm 10.9$ | $56.1 \pm 6.1$ | $57.8 \pm 2.3$ | $58.6 \pm 0.9$ | $58.9 \pm 0.7$ | $59.0 \pm 0.7$ | $59.1 \pm 0.6$ |
| | **ReBRAC+CE+AT** | $43.9 \pm 17.1$ | $52.7 \pm 10.3$ | $55.6 \pm 6.0$ | $57.5 \pm 2.5$ | $58.5 \pm 0.9$ | $58.7 \pm 0.6$ | $58.8 \pm 0.6$ | $58.9 \pm 0.5$ |
| | **ReBRAC+CE+CT** | $56.9 \pm 1.7$ | $57.9 \pm 1.3$ | $58.4 \pm 1.0$ | $58.8 \pm 0.7$ | $59.2 \pm 0.4$ | $59.3 \pm 0.3$ | $59.4 \pm 0.3$ | $59.4 \pm 0.2$ |
| | **ReBRAC+CE+MT** | $40.8 \pm 17.0$ | $49.8 \pm 12.0$ | $53.5 \pm 8.1$ | $56.3 \pm 4.1$ | $57.8 \pm 1.4$ | $58.0 \pm 0.6$ | $58.1 \pm 0.4$ | - |
| | **IQL** | $33.9 \pm 2.3$ | $35.2 \pm 1.7$ | $35.8 \pm 1.5$ | $36.4 \pm 1.2$ | $37.1 \pm 0.9$ | $37.4 \pm 0.6$ | $37.5 \pm 0.6$ | $37.5 \pm 0.5$ |
| | **IQL+CE+AT** | $53.0 \pm 3.5$ | $55.0 \pm 3.1$ | $56.1 \pm 2.7$ | $57.2 \pm 1.9$ | $58.1 \pm 0.8$ | $58.3 \pm 0.4$ | $58.4 \pm 0.3$ | $58.4 \pm 0.3$ |
| | **IQL+CE+CT** | $49.6 \pm 1.3$ | $50.4 \pm 0.9$ | $50.7 \pm 0.7$ | $51.0 \pm 0.6$ | $51.3 \pm 0.4$ | $51.4 \pm 0.4$ | $51.5 \pm 0.3$ | $51.5 \pm 0.3$ |
| | **IQL+CE+MT** | $54.2 \pm 3.1$ | $55.9 \pm 2.3$ | $56.7 \pm 1.7$ | $57.4 \pm 1.0$ | $57.8 \pm 0.3$ | $57.8 \pm 0.1$ | $57.8 \pm 0.1$ | - |
| | **LB-SAC** | $15.7 \pm 14.1$ | $23.4 \pm 12.8$ | $27.7 \pm 10.5$ | $31.7 \pm 6.6$ | - | - | - | - |
| | **LB-SAC+CE+AT** | $12.7 \pm 8.0$ | $17.1 \pm 5.9$ | $19.1 \pm 4.3$ | $20.6 \pm 2.3$ | - | - | - | - |
| | **LB-SAC+CE+CT** | $22.0 \pm 2.7$ | $23.4 \pm 2.6$ | $24.2 \pm 2.7$ | $25.2 \pm 2.6$ | $26.4 \pm 2.2$ | - | - | - |
| | **LB-SAC+CE+MT** | $12.5 \pm 8.8$ | $17.3 \pm 6.5$ | $19.4 \pm 4.6$ | $21.1 \pm 2.6$ | $22.3 \pm 1.5$ | $22.8 \pm 1.2$ | $23.0 \pm 1.1$ | - |

We present algorithms' performance heatmaps for the parameter grid averaged over domains in Figure 1, with detailed scores per dataset available in Appendix J. Additionally, in Appendix F, we provide further insights by fixing one parameter and averaging over the second. Our analysis indicates that setting $m = 101$ and $\sigma/\zeta = 0.75$ may serve as a reasonable starting point and produce results better than the average. Notably,

---

[3] 9 policies for LB-SAC+CE+CT.

a higher number of classes often yields improved performance, suggesting a preference for larger $m$ values. Regarding $\sigma/\zeta$, our findings suggest that 0.75 remains a favorable choice, especially when faced with limited tuning resources. However, the optimal selection of classification parameters heavily depends on the specific algorithm, environment, and dataset characteristics.

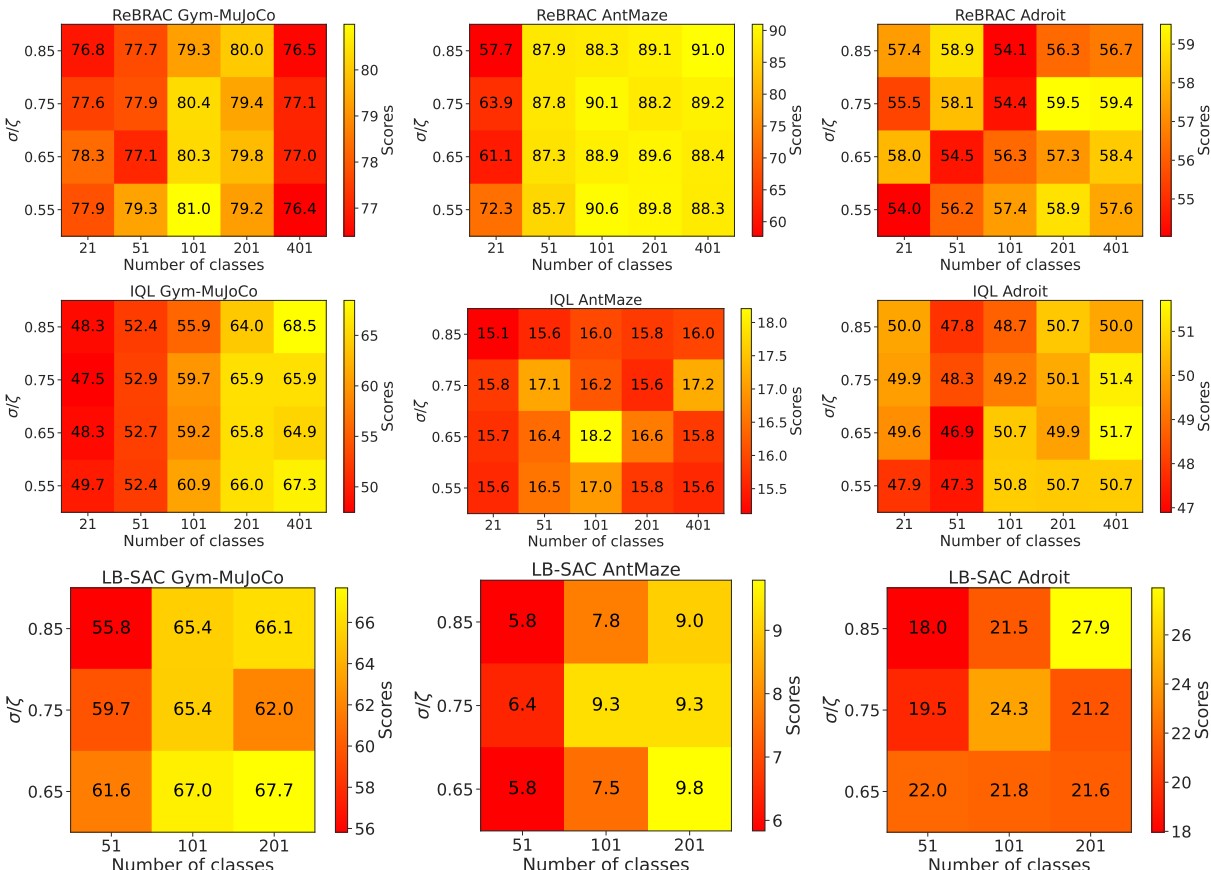

Figure 1: Heatmaps for the impact of the classification parameters averaged over the domains. See Appendix J and Appendix F for more results.

Farebrother et al. (2024) did not investigate the impact of the $v_{min}$ and $v_{max}$ choices, nor did they propose a method for selecting these when the reward function is unknown. In our previous experiments, we computed these values by taking the minimum and maximum return across all possible sub-trajectories in the dataset. However, this approach may not be optimal because offline RL algorithms often introduce pessimism into the Q function. Additionally, setting these limits based on extreme values may cause the values on the edges of the support to differ from other values from the model's perspective, potentially impacting the final results.

To evaluate the effect of these parameters, we computed the support size as $v_{max} - v_{min}$ from the dataset and multiplied this size by a parameter $v_{expand}$. We then extended the support in two ways: Subtracting $v_{expand}(v_{max} - v_{min})$ from $v_{min}$ (referred to as $min$), and simultaneously subtracting from $v_{min}$ and adding to $v_{max}$ the term $v_{expand}(v_{max} - v_{min})/2$ (referred to as $both$)[4]. The experimental results, shown in Figure 8, demonstrate that our initial choice of support parameters was sub-optimal and increasing the support range may strongly benefit. Surprisingly, in some cases, reducing the support was beneficial. The $both$ strategy generally yielded better performance.

Our observations support the assumption about efficiency of cross-entropy for offline RL algorithms with policy regularization compared to other types. Notably, ReBRAC exhibited less sensitivity to the classification parameters.

---

[4]The division by 2 ensures equal bin sizes across the two variants.

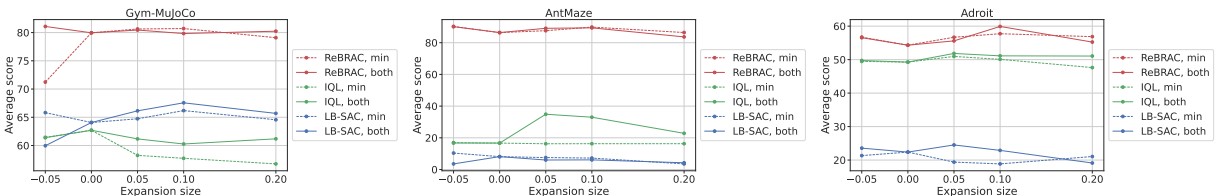

Figure 2: Dependency of the algorithms performance on $v_{expand}$ values averaged over domains. See Appendix G for tabular representation.

### 4.5 Do MLPs scale better with classification?

In Farebrother et al. (2024), authors emphasized the enhanced scalability of models when regression is replaced with classification, particularly when using Transformers or ResNets. In this subsection, we delve into whether similar scalability benefits apply to Multilayer Perceptrons (MLPs), which are commonly utilized in the development of novel RL approaches. Experimental results are presented in Figure 3. Surprisingly, our findings suggest that there is no consistent improvement in terms of scaling when MSE is replaced with cross-entropy.

This discrepancy from the results reported by Farebrother et al. (2024) could be attributed to the fundamental architectural differences between MLPs and models like Transformers or ResNets. Unlike these latter architectures, vanilla MLPs typically lack residual connections (He et al., 2016), which have been identified as a crucial factor in enabling effective scaling with depth. Consequently, the absence of such connections in MLPs may limit their ability to capitalize on the benefits offered by classification over regression.

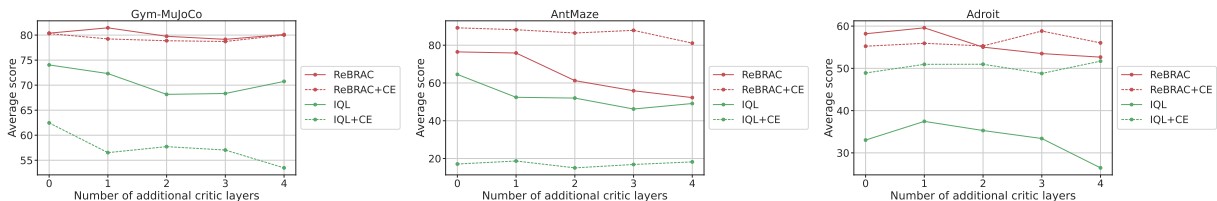

Figure 3: Dependency of the algorithms performance on the number of additional layers averaged over domains. See Appendix E for tabular representation.

### 4.6 Combining the findings

We investigated whether a mixed tuning strategy for hyperparameters could enhance classification performance, building on our previous findings. For each algorithm, we tuned the following parameters: $m$ from the set $\{201, 401\}$, $v_{expand}$ from the set of $\{-0.05, 0.05, 0.1\}$ using the *both* strategy, and three values for one of algorithm-specific parameter: $\beta_1$ for ReBRAC, IQL $\tau$ for IQL, and N critics for LB-SAC. For ReBRAC and IQL, the second parameter was held constant across all tasks (see subsection A.1). $\sigma/\zeta$ was set to 0.75. To enhance readability, we provide detailed per-dataset results in a separate tables in Appendix H and EOP results in Table 4 under the **+CE+MT** rows (see also Appendix I).

Per-dataset results indicate that the proposed strategy yields better performance in most cases, performing on par in others, with the exception of LB-SAC on Gym-MuJoCo. It makes this approach the best choice when no prior knowledge on optimal algorithm-specific parameters is available.

EOP results reveal that this tuning strategy is particularly effective for IQL, especially under a low fine-tuning budget. For ReBRAC, benefits are apparent only under a high fine-tuning budget. Conversely, this strategy did not perform well for LB-SAC. The suboptimal performance of LB-SAC can be attributed to the strong dependence on ensemble size; due to computational constraints, we were unable to use an ensemble size of 50, which is optimal for many datasets.

To conclude, our experiments show that the best performance with classification can be achieved by tuning algorithm-specific hyperparameters in conjunction with classification parameters, particularly the support range and the number of bins.

## 4.7 Learned Q-functions analysis

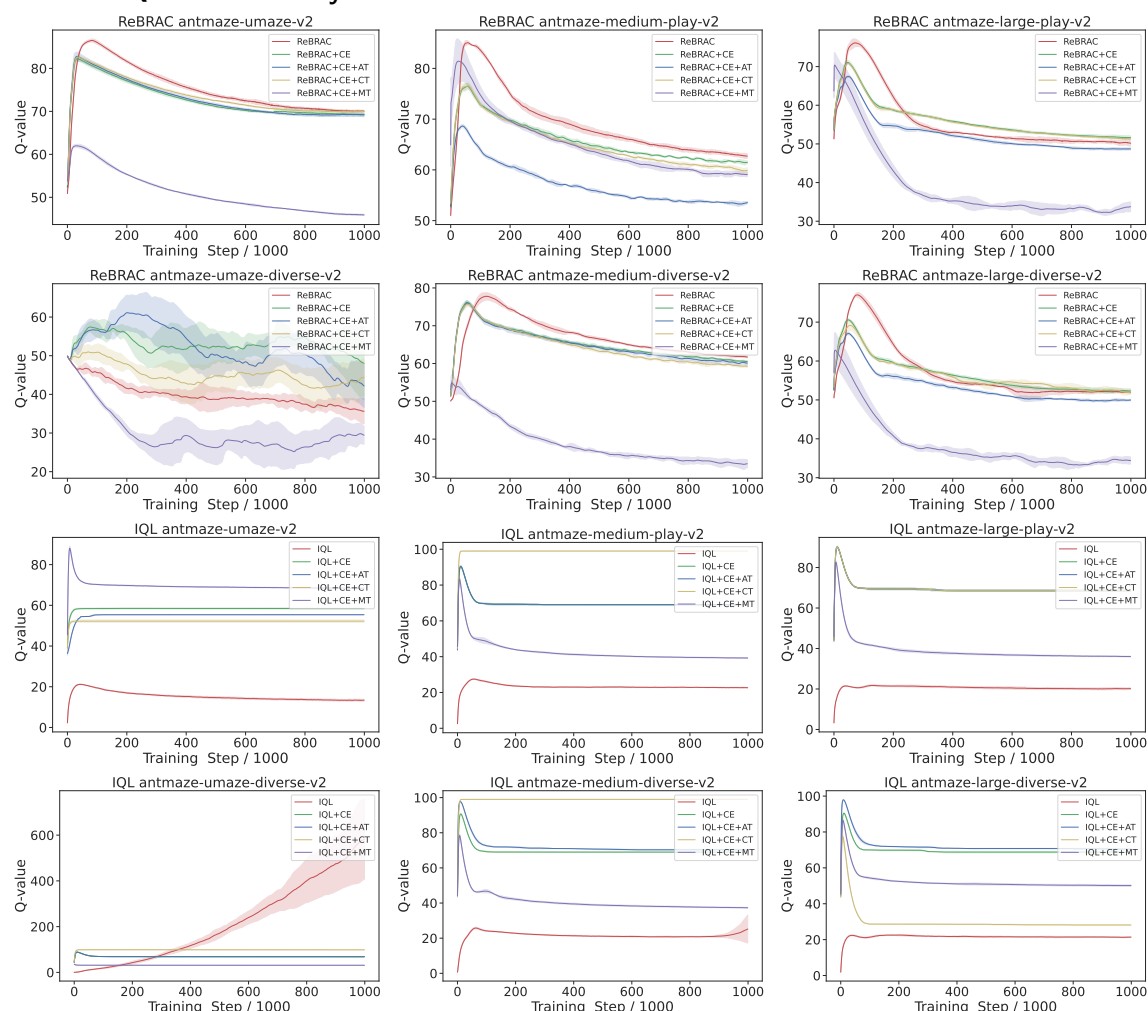

Figure 4: Q-value functions behaviour for ReBRAC and IQL on AntMaze tasks. Shaded area demonstrates standard deviation across ten random seeds.

Finally, in Figure 4 we present the dynamics of the Q-function learning for ReBRAC and IQL on AntMaze datasets, as they are of the most interest based on our results. For ReBRAC, modified versions have lower Q-function values on average, but the learning pattern remains similar, except for **CE+MT**, where the Q-values are notably lower, and the shape of the curve is different. However, these differences in the Q-function between **CE+MT** and **CE+AT**/**CE+CT** do not result in significant differences in average performance. For IQL, the plots demonstrate that the usage of classification leads to more optimistic Q-functions, with a significant gap between MSE and others. Unfortunately, there is no consistent pattern. We cannot claim that a more optimistic Q-function is the core issue for IQL because, for example, **CE+MT** is sometimes above **CE+CT**, but this has no correlation with task performance.

## 5 Related Work

Prior research beyond the realm of RL has demonstrated the potential performance enhancements associated with replacing regression with classification objectives (Van Den Oord et al., 2016; Kendall et al., 2017;

Rothe et al., 2018; Rogez et al., 2017). Within the RL domain, some studies have experimented with employing classification objectives as a workaround, albeit without conducting comprehensive analyses of this modification (Schrittwieser et al., 2020; Hafner et al., 2023; Hessel et al., 2021; Hansen et al., 2023). Categorical distributional RL (Bellemare et al., 2017) works are also relevant for the considered topic, where the classification is also used, however usage of classification instead of regression is not a central topic in this reseach direction. Additionally, several works in offline RL have demonstrated the benefits of utilizing classification objectives for various tasks, albeit lacking in-depth analyses of this specific component and its elements (Kumar et al., 2022; Springenberg et al., 2024).

To the best of our knowledge, (Farebrother et al., 2024) represent the first and only study to make regression replacement with classification a central research question. The authors compared different methods of converting regression targets into classification targets, providing experimental results across a diverse array of tasks encompassing Atari games, robotics, and natural language processing problems in online and offline RL setups. Their study revealed that HL-Gauss (Imani & White, 2018) represents the optimal approach for representing RL regression targets with categorical distributions while this was not the case for the supervised regression tasks (Imani & White, 2018). The authors assert that cross-entropy serves as a "drop-in" replacement for MSE in RL, leading to a more stable training process and enhanced scalability with deep neural network architectures such as Transformers (Vaswani et al., 2017) or ResNets (He et al., 2016). Our work draws primary inspiration from (Farebrother et al., 2024) and aims to provide a more in-depth analysis of this phenomenon in offline RL, which we believe holds significant potential benefits for both offline RL researchers and practitioners.

## 6 Limitations and Future Work

Looking ahead, several avenues for future research emerge. Firstly, one promising direction involves examining how classification affects the performance of offline algorithms in offline-to-online RL setup. Understanding how classification objectives impact the transferability of learned policies to online settings could provide valuable insights into the practical applicability of classification usage. One limitation here, however, is that dataset-specific finite return ranges may restrict policy generalization when transitioning to online tuning. This could require new design choices to handle potentially expanded return ranges in online scenarios.

Second, our approach applies a relatively straightforward replacement of regression with classification, and future studies could explore more refined adaptations. Although this naive approach fits naturally for ReBRAC and LB-SAC, for IQL we replaced the MSE with cross-entropy only in the Q-network, retaining the expectile loss for the V-function. Expectile loss, as a more general form of MSE, is integral to IQL's performance, and investigating whether it could be effectively adapted to a classification-based objective is an exciting direction. However, such a replacement is mathematically complex and requires additional research. For algorithms like ReBRAC and LB-SAC, which incorporate pessimism in the Q-function, it may also be worth exploring whether discretization should occur before or after the pessimism adjustment, rather than afterward as in our study.

Our findings also offer practical insights for the development of new offline RL algorithms. For example, our results with ReBRAC suggest that classification-based objectives could mitigate Q-function divergence, a recurring challenge in off-policy and offline RL (Van Hasselt et al., 2018). Moreover, the guidance we provide on hyperparameter choices for classification losses could assist RL practitioners and researchers in fine-tuning these algorithms for better performance if they decide to incorporate classification loss. For instance, a recent modification of ReBRAC by Tarasov et al. (2024c) is based on our modification of it, leading to near-resolution of the AntMaze domain in the D4RL benchmark suite.

Another theoretically promising direction is leveraging classification-specifics for uncertainty estimation in offline RL. For example, the entropy provided by the Q function could be used to incorporate pessimism into offline RL algorithms.

# 7 Conclusion

In this study, we explored the impact of integrating classification objectives into offline RL algorithms. Our findings provide nuanced insights into the efficacy of classification in improving offline RL algorithm performance. Initially, we examined whether classification objectives could be seamlessly integrated into existing algorithms without altering other aspects. While some algorithm with policy regularization (ReBRAC) demonstrated promising results across various tasks , other algorithms with implicit regularization (IQL) and algorithm with Q function regularization (LB-SAC) faced challenges. And the only case when classification have high chances to bring improvement without much effort is the divergence of Q function with MSE loss.

Next, we explored the impact of different hyperparameter tuning with classification objectives. Notably, we observed performance improvements for ReBRAC, when tuning classification hyperparameters over algorithm-specific ones. Other results underscore the importance of carefully selecting both types of hyperparameters: algorithm-specific and classification-specific, when employing classification.

Furthermore, our investigation into the scalability of MLPs with classification revealed mixed results. Contrary to previous findings with architectures like Transformers and ResNets, we did not observe consistent improvements in scaling when using classification objectives with MLPs. This highlights the importance of considering the architectural nuances of different models when assessing the potential benefits of classification objectives.

In conclusion, our study underscores the need for a nuanced understanding of the interplay between algorithm design, task characteristics, and the integration of classification objectives in RL. While classification holds promise in certain contexts, its efficacy is highly dependent on factors such as algorithm design and hyperparameter selection. Future research could further explore these nuances and develop approaches that leverage classification objectives optimally across a diverse range of RL tasks and algorithms.

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

# A    Experimental Details

For results with original algorithms and algorithms hyperparameters search in Table 1 Table 2 and Table 3, we conducted a hyperparameter search and selected the best results from the final evaluations for each dataset. We used the JAX implementation of ReBRAC from the Clean Offline RL (CORL) library (Tarasov et al., 2024b) and used the same code template for IQL and LB-SAC. Algorithmic part of IQL implementation is based on the original codebase from Kostrikov et al. (2021) and in case of LB-SAC we have adapted SAC-N implementation from `https://github.com/Howuhh/sac-n-jax`.

The experiments were conducted on RTX Titan, Quadro RTX 6000 and Quadro RTX 8000.

Our study utilized the v2 version of datasets for Gym-MuJoCo and AntMaze, and v1 for Adroit. The agents were trained for one million steps in all domains and evaluated over ten episodes for Gym-MuJoCo and Adroit and over one hundred episodes for AntMaze. Following Chen et al. (2022), AntMaze reward function is multiplied by 100.

For ReBRAC, we fine-tuned the $\beta_1$ parameter with $0.001, 0.01, 0.05, 0.1$ values and the $\beta_2$ parameter with $0, 0.001, 0.01, 0.1, 0.5$ values for Gym-MuJoCo and Adroit. For AntMaze the corresponding ranges are $0.0005, 0.001, 0.002, 0.003$ and $0, 0.0001, 0.0005, 0.001$. When replacing regression with classification for AntMaze we also set batch size to 1024 and learning rates to 0.001 as it is was done for Gym-MuJoCo in original algorithm which slightly improves the performance.

For IQL in all domains, we selected $\beta$ value from $0.5, 1, 3, 6, 10$ and IQL $\tau$ from $0.5, 0.7, 0.9, 0.95$.

For LB-SAC we selected the number of critics in the range of $2, 5, 10, 25, 50$. Note, that for LB-SAC we used sub-optimal parameters of the batch size and learning rate due to the computational constraints.

## A.1    Mixed tuning hyperparameters choice

For ReBRAC, we fine-tuned the $\beta_1$ parameter with $0.001, 0.01, 0.05$ values and the $\beta_2$ parameter was set to 0.001.

For IQL, we selected IQL $\tau$ from $0.5, 0.7, 0.9$ and set $\beta$ value to 3.

For LB-SAC we selected the number of critics in the range of $2, 10, 25$.

## B    Algorithms Pseudocodes

Listing 1: Util functions used in our implementations in Python JAX.

```python
def transform_to_probs(target: jax.Array, support: jax.Array, sigma: float) -> jax.Array:
    cdf_evals = jax.scipy.special.erf((support - target) / (jnp.sqrt(2) * sigma))
    z = cdf_evals[-1] - cdf_evals[0]
    bin_probs = cdf_evals[1:] - cdf_evals[:-1]
    return bin_probs / (z + 1e-6)
# For batch processing
transform_to_probs = jax.vmap(transform_to_probs, in_axes=(0, None, None))

def transform_from_probs(probs: jax.Array, support: jax.Array) -> jax.Array:
    centers = (support[:-1] + support[1:]) / 2
    return jnp.sum(probs * centers)
# For batch processing
transform_from_probs = jax.vmap(transform_from_probs, in_axes=(0, None))
transform_from_probs = jax.vmap(transform_from_probs, in_axes=(0, None))
```

Listing 2: ReBRAC actor and critic updates with integrated cross-entropy Python JAX pseudocode.

```python
def update_actor(
    key: jax.random.PRNGKey, actor: TrainState, critic: TrainState,
    batch: Dict[str, jax.Array], beta: float, tau: float, normalize_q: bool,
):
    def actor_loss_fn(params: jax.Array):
        actions = actor.apply_fn(params, batch["states"])
        bc_penalty = ((actions - batch["actions"]) ** 2).sum(-1)
        # Predict Q categorical distributions
        logits = critic.apply_fn(critic.params, batch["states"], actions)
        probs = nn.softmax(logits, axis=-1)
        # Convert distributions to scalar values
        q_values = transform_from_probs(probs, critic.support).min(0)
        lmbda = 1
        if normalize_q:
            lmbda = jax.lax.stop_gradient(1 / jax.numpy.abs(q_values).mean())
        loss = (beta * bc_penalty - lmbda * q_values).mean()
        return loss

    grads = jax.grad(actor_loss_fn)(actor.params)
    new_actor = actor.apply_gradients(grads=grads)
    new_actor = new_actor.replace(
        target_params=optax.incremental_update(actor.params, actor.target_params, tau)
    )
    new_critic = critic.replace(
        target_params=optax.incremental_update(critic.params, critic.target_params, tau)
    )
    return key, new_actor, new_critic

def update_critic(
    key: jax.random.PRNGKey, actor: TrainState, critic: CriticTrainState,
    batch: Dict[str, jax.Array], gamma: float, beta: float, policy_noise: float, noise_clip:
        float,
):
    key, actions_key = jax.random.split(key)

    next_actions = actor.apply_fn(actor.target_params, batch["next_states"])
    noise = jax.numpy.clip(
        (jax.random.normal(actions_key, next_actions.shape) * policy_noise),
        -noise_clip,
        noise_clip,
    )
    next_actions = jax.numpy.clip(next_actions + noise, -1, 1)
    bc_penalty = ((next_actions - batch["next_actions"]) ** 2).sum(-1)
    # Predict target Q categorical distributions
    logits = critic.apply_fn(critic.target_params, batch["next_states"], next_actions)
    probs = nn.softmax(logits, axis=-1)
    # Convert distributions to target scalar values
    next_q = transform_from_probs(probs, critic.support).min(0)
    next_q = next_q - beta * bc_penalty
    target_q = batch["rewards"] + (1 - batch["dones"]) * gamma * next_q

    def critic_loss_fn(critic_params: jax.Array):
        # Predict Q categorical distributions
        q = critic.apply_fn(critic_params, batch["states"], batch["actions"])
        # Convert target Q values to categorical distribution
        target_probs = transform_to_probs(target_q, critic.support, critic.sigma)
        # Compute CE loss
        loss = optax.softmax_cross_entropy(logits=q, labels=target_probs[None, ...]).mean(1)
            .sum(0)
        return loss

    loss, grads = jax.value_and_grad(critic_loss_fn)(critic.params)
    new_critic = critic.apply_gradients(grads=grads)
    return key, new_critic
```

Listing 3: IQL actor, V and Q updates with integrated cross-entropy Python JAX pseudocode.

```python
def update_actor(
    key: jax.random.PRNGKey, actor: TrainState, critic: CriticTrainState, value: TrainState,
    batch: Dict[str, Any], temperature: float,
):
    key, random_dropout_key = jax.random.split(key, 2)

    v = value.apply_fn(value.params, batch["states"])
    # Predict Q categorical distributions
    logits1, logits2 = critic.apply_fn(critic.target_params, batch["states"], batch["actions"])
    probs1, probs2 = nn.softmax(logits1, axis=-1), nn.softmax(logits2, axis=-1)
    # Convert distributions to scalar values
    q1, q2 = transform_from_probs(probs1, critic.support), transform_from_probs(probs2,
        critic.support)
    q = jnp.minimum(q1, q2)
    exp_a = jnp.exp((q - v) * temperature)
    exp_a = jnp.clip(exp_a, -100.0, 100.0)

    def actor_loss_fn(actor_params):
        dist = actor.apply_fn(actor_params, batch["states"], training=True, rngs={'dropout':
            random_dropout_key})
        eps = 1e-6
        log_probs = dist.log_prob(jax.numpy.clip(batch["actions"], -1 + eps, 1 - eps))
        actor_loss = -(exp_a * log_probs).mean()
        return actor_loss

    loss, grads = jax.value_and_grad(actor_loss_fn)(actor.params)
    new_actor = actor.apply_gradients(grads=grads)
    return key, new_actor

def update_v(
    key: jax.random.PRNGKey, critic: CriticTrainState, value: TrainState,
    batch: Dict[str, Any], expectile: float
):
    # Predict Q categorical distributions
    logits1, logits2 = critic.apply_fn(critic.target_params, batch["states"], batch["actions"])
    probs1, probs2 = nn.softmax(logits1, axis=-1), nn.softmax(logits2, axis=-1)
    # Convert distributions to scalar values
    q1, q2 = transform_from_probs(probs1, critic.support), transform_from_probs(probs2,
        critic.support)
    q = jnp.minimum(q1, q2)

    def expectile_loss(diff, expectile=0.8):
        weight = jnp.where(diff > 0, expectile, (1 - expectile))
        return weight * (diff ** 2)

    def value_loss_fn(value_params):
        v = value.apply_fn(value_params, batch["states"])
        value_loss = expectile_loss(q - v, expectile).mean()
        return value_loss

    loss, grads = jax.value_and_grad(value_loss_fn)(value.params)
    new_value = value.apply_gradients(grads=grads)
    return key, new_value

def update_q(
        key: jax.random.PRNGKey, critic: CriticTrainState, target_value: TrainState,
        batch: Dict[str, Any], gamma: float,
):
    next_v = target_value.apply_fn(target_value.params, batch["next_states"])
    target_q = batch["rewards"] + gamma * (1 - batch["dones"]) * next_v
    # Convert target Q values to categorical distribution
    target_probs = transform_to_probs(target_q, critic.support, critic.sigma)
```

```
62      def critic_loss_fn(critic_params):
63          # Predict Q categorical distributions
64          q1, q2 = critic.apply_fn(critic_params, batch["states"], batch["actions"])
65          # Compute CE losses
66          critic_loss1 = optax.softmax_cross_entropy(logits=q1, labels=target_probs[None,
                ...]).mean(1).sum(0)
67          critic_loss2 = optax.softmax_cross_entropy(logits=q2, labels=target_probs[None,
                ...]).mean(1).sum(0)
68          critic_loss = critic_loss1 + critic_loss2
69          return critic_loss
70
71      loss, grads = jax.value_and_grad(critic_loss_fn)(critic.params)
72      new_critic = critic.apply_gradients(grads=grads)
73      return key, new_critic
```

Listing 4: LB-SAC actor and critic updates with integrated cross-entropy Python JAX pseudocode.

```
1   def update_actor(
2       key: jax.random.PRNGKey, actor: TrainState, critic: TrainState, alpha: TrainState,
3       batch: Dict[str, jax.Array]
4   ):
5       def actor_loss_fn(actor_params):
6           actions_dist = actor.apply_fn(actor_params, batch["states"])
7           actions, actions_logp = actions_dist.sample_and_log_prob(seed=key)
8           # Predict Q categorical distributions
9           logits = critic.apply_fn(critic.params, batch["states"], actions)
10          probs = nn.softmax(logits, axis=-1)
11          # Convert distributions to scalar values
12          q_values = transform_from_probs(probs, critic.support).min(0)
13          loss = (alpha.apply_fn(alpha.params) * actions_logp.sum(-1) - q_values).mean()
14          return loss
15
16      loss, grads = jax.value_and_grad(actor_loss_fn)(actor.params)
17      new_actor = actor.apply_gradients(grads=grads)
18      return new_actor
19
20  def update_critic(
21      key: jax.random.PRNGKey, actor: TrainState, critic: CriticTrainState, alpha: TrainState,
22      batch: Dict[str, jax.Array], gamma: float, tau: float,
23  ):
24      next_actions_dist = actor.apply_fn(actor.params, batch["next_states"])
25      next_actions, next_actions_logp = next_actions_dist.sample_and_log_prob(seed=key)
26      # Predict target Q categorical distributions
27      logits = critic.apply_fn(critic.target_params, batch["next_states"], next_actions)
28      probs = nn.softmax(logits, axis=-1)
29      # Convert distributions to target scalar values
30      next_q = transform_from_probs(probs, critic.support).min(0)
31      next_q = next_q - alpha.apply_fn(alpha.params) * next_actions_logp.sum(-1)
32      target_q = batch["rewards"] + (1 - batch["dones"]) * gamma * next_q
33
34      def critic_loss_fn(critic_params):
35          # Predict Q categorical distributions
36          q = critic.apply_fn(critic_params, batch["states"], batch["actions"])
37          # Convert target Q values to categorical distribution
38          target_probs = transform_to_probs(target_q, critic.support, critic.sigma)
39          # Compute CE loss
40          loss = optax.softmax_cross_entropy(logits=q, labels=target_probs[None, ...]).mean(1)
                .sum(0)
41          return loss
42
43      loss, grads = jax.value_and_grad(critic_loss_fn)(critic.params)
44      new_critic = critic.apply_gradients(grads=grads).soft_update(tau=tau)
45      return new_critic
```

# C Hyperparameters

## C.1 ReBRAC

Table 5: ReBRAC's general hyperparameters.

| Parameter | Value |
|---|---|
| optimizer | Adam Kingma & Ba (2014) |
| batch size | 1024 on Gym-MuJoCo and AntMaze (for classification), 256 otherwise |
| learning rate (all networks) | 1e-3 on Gym-MuJoCo and AntMaze (for classification), 3e-4 otherwise |
| tau ($\tau$) | 5e-3 |
| hidden dim (all networks) | 256 |
| num hidden layers (all networks) | 3 |
| gamma ($\gamma$) | 0.999 on AntMaze, 0.99 otherwise |
| nonlinearity | ReLU |

Table 6: ReBRAC's best hyperparameters.

| Task Name | $\beta_1$ (MSE) | $\beta_2$ (MSE) | $\beta_1$ (CE+AT) | $\beta_2$ (CE+AT) | $m$ (CE+CT) | $\sigma/\zeta$ (CE+CT) | $\beta_1$ (CE+MT) | $m$ (CE+MT) | $v_{expand}$ (CE+MT) |
|---|---|---|---|---|---|---|---|---|---|
| halfcheetah-random | 0.001 | 0.1 | 0.001 | 0.1 | 101 | 0.85 | 0.001 | 201 | 0.05 |
| halfcheetah-medium | 0.001 | 0.01 | 0.001 | 0.5 | 21 | 0.75 | 0.001 | 201 | 0.1 |
| halfcheetah-expert | 0.01 | 0.01 | 0.01 | 0.01 | 201 | 0.85 | 0.01 | 201 | 0.05 |
| halfcheetah-medium-expert | 0.01 | 0.1 | 0.01 | 0.001 | 101 | 0.85 | 0.01 | 201 | 0.1 |
| halfcheetah-medium-replay | 0.01 | 0.001 | 0.001 | 0.001 | 201 | 0.65 | 0.001 | 201 | 0.1 |
| halfcheetah-full-replay | 0.001 | 0.1 | 0.001 | 0.001 | 21 | 0.75 | 0.001 | 201 | -0.05 |
| hopper-random | 0.001 | 0.01 | 0.001 | 0.5 | 201 | 0.85 | 0.001 | 401 | -0.05 |
| hopper-medium | 0.01 | 0.001 | 0.01 | 0.001 | 201 | 0.75 | 0.01 | 201 | -0.05 |
| hopper-expert | 0.1 | 0.001 | 0.1 | 0.1 | 401 | 0.65 | 0.05 | 401 | 0.1 |
| hopper-medium-expert | 0.1 | 0.01 | 0.1 | 0.0 | 101 | 0.75 | 0.05 | 401 | 0.05 |
| hopper-medium-replay | 0.05 | 0.5 | 0.05 | 0.0 | 101 | 0.75 | 0.01 | 401 | -0.05 |
| hopper-full-replay | 0.01 | 0.01 | 0.01 | 0.0 | 101 | 0.65 | 0.01 | 201 | 0.05 |
| walker2d-random | 0.01 | 0.0 | 0.05 | 0.1 | 101 | 0.65 | 0.05 | 201 | 0.05 |
| walker2d-medium | 0.05 | 0.1 | 0.05 | 0.1 | 101 | 0.55 | 0.05 | 201 | 0.1 |
| walker2d-expert | 0.01 | 0.5 | 0.01 | 0.5 | 101 | 0.75 | 0.01 | 401 | 0.05 |
| walker2d-medium-expert | 0.01 | 0.01 | 0.01 | 0.5 | 401 | 0.85 | 0.05 | 401 | 0.05 |
| walker2d-medium-replay | 0.05 | 0.01 | 0.05 | 0.0 | 201 | 0.65 | 0.01 | 401 | -0.05 |
| walker2d-full-replay | 0.01 | 0.01 | 0.001 | 0.001 | 201 | 0.55 | 0.001 | 201 | 0.1 |
| antmaze-umaze | 0.003 | 0.002 | 0.003 | 0.0 | 401 | 0.75 | 0.05 | 401 | 0.1 |
| antmaze-umaze-diverse | 0.003 | 0.001 | 0.003 | 0.002 | 201 | 0.75 | 0.01 | 201 | 0.1 |
| antmaze-medium-play | 0.001 | 0.0005 | 0.003 | 0.0005 | 201 | 0.65 | 0.001 | 401 | 0.05 |
| antmaze-medium-diverse | 0.001 | 0.0 | 0.003 | 0.002 | 201 | 0.85 | 0.01 | 401 | 0.1 |
| antmaze-large-play | 0.002 | 0.001 | 0.003 | 0.0005 | 101 | 0.75 | 0.01 | 401 | 0.1 |
| antmaze-large-diverse | 0.002 | 0.002 | 0.003 | 0.0005 | 401 | 0.75 | 0.01 | 401 | 0.1 |
| pen-human | 0.1 | 0.5 | 0.1 | 0.001 | 51 | 0.85 | 0.05 | 201 | -0.05 |
| pen-cloned | 0.05 | 0.5 | 0.1 | 0.001 | 401 | 0.75 | 0.05 | 401 | 0.05 |
| pen-expert | 0.01 | 0.01 | 0.01 | 0.0 | 401 | 0.85 | 0.01 | 401 | -0.05 |
| door-human | 0.1 | 0.1 | 0.1 | 0.1 | 401 | 0.85 | 0.001 | 401 | -0.05 |
| door-cloned | 0.01 | 0.1 | 0.1 | 0.001 | 101 | 0.55 | 0.05 | 201 | 0.1 |
| door-expert | 0.05 | 0.01 | 0.05 | 0.001 | 401 | 0.75 | 0.05 | 401 | -0.05 |
| hammer-human | 0.01 | 0.5 | 0.1 | 0.001 | 51 | 0.85 | 0.05 | 201 | -0.05 |
| hammer-cloned | 0.1 | 0.5 | 0.05 | 0.1 | 21 | 0.85 | 0.05 | 401 | 0.05 |
| hammer-expert | 0.01 | 0.01 | 0.01 | 0.5 | 401 | 0.55 | 0.01 | 401 | 0.1 |
| relocate-human | 0.1 | 0.01 | 0.1 | 0.01 | 201 | 0.55 | 0.05 | 201 | 0.05 |
| relocate-cloned | 0.1 | 0.01 | 0.1 | 0.0 | 401 | 0.75 | 0.05 | 201 | 0.1 |
| relocate-expert | 0.05 | 0.01 | 0.05 | 0.5 | 21 | 0.55 | 0.05 | 401 | 0.05 |

## C.2 IQL

Table 7: IQL's general hyperparameters.

| Parameter | Value |
|---|---|
| optimizer | Adam Kingma & Ba (2014) |
| batch size | 256 |
| learning rate (all networks) | 3e-4 |
| tau ($\tau$) | 5e-3 |
| hidden dim (all networks) | 256 |
| num hidden layers (all networks) | 2 |
| gamma ($\gamma$) | 0.99 |
| nonlinearity | ReLU |
| learning rate decay | Cosine |
| dropout rate | 0.1 for Adroit, 0 otherwise |

Table 8: IQL's best hyperparameters.

| Task Name | IQL $\tau$ (MSE) | $\beta$ (MSE) | IQL $\tau$ (CE+AT) | $\beta$ (CE+AT) | $m$ (CE+CT) | $\sigma/\zeta$ (CE+CT) | IQL $\tau$ (CE+MT) | $m$ (CE+MT) | $v_{expand}$ (CE+MT) |
|---|---|---|---|---|---|---|---|---|---|
| halfcheetah-random | 0.95 | 10.0 | 0.5 | 3.0 | 401 | 0.55 | 0.7 | 401 | -0.05 |
| halfcheetah-medium | 0.95 | 3.0 | 0.5 | 10.0 | 401 | 0.55 | 0.7 | 401 | 0.1 |
| halfcheetah-expert | 0.7 | 6.0 | 0.7 | 1.0 | 401 | 0.55 | 0.5 | 401 | 0.1 |
| halfcheetah-medium-expert | 0.5 | 0.5 | 0.5 | 3.0 | 401 | 0.75 | 0.7 | 401 | 0.1 |
| halfcheetah-medium-replay | 0.9 | 6.0 | 0.5 | 6.0 | 401 | 0.55 | 0.7 | 401 | 0.1 |
| halfcheetah-full-replay | 0.5 | 0.5 | 0.5 | 6.0 | 401 | 0.55 | 0.5 | 401 | 0.1 |
| hopper-random | 0.95 | 10.0 | 0.95 | 6.0 | 51 | 0.85 | 0.9 | 201 | -0.05 |
| hopper-medium | 0.7 | 0.5 | 0.7 | 10.0 | 51 | 0.85 | 0.5 | 201 | 0.05 |
| hopper-expert | 0.9 | 0.5 | 0.7 | 3.0 | 201 | 0.75 | 0.9 | 201 | -0.05 |
| hopper-medium-expert | 0.7 | 10.0 | 0.5 | 3.0 | 201 | 0.85 | 0.5 | 401 | 0.05 |
| hopper-medium-replay | 0.7 | 0.5 | 0.5 | 6.0 | 401 | 0.85 | 0.7 | 401 | 0.05 |
| hopper-full-replay | 0.7 | 3.0 | 0.5 | 3.0 | 401 | 0.55 | 0.5 | 401 | -0.05 |
| walker2d-random | 0.9 | 0.5 | 0.95 | 3.0 | 21 | 0.65 | 0.9 | 401 | 0.1 |
| walker2d-medium | 0.5 | 1.0 | 0.5 | 0.5 | 101 | 0.55 | 0.7 | 401 | 0.1 |
| walker2d-expert | 0.7 | 6.0 | 0.7 | 10.0 | 401 | 0.55 | 0.7 | 401 | 0.05 |
| walker2d-medium-expert | 0.7 | 3.0 | 0.5 | 6.0 | 401 | 0.85 | 0.5 | 401 | 0.1 |
| walker2d-medium-replay | 0.7 | 1.0 | 0.5 | 6.0 | 401 | 0.75 | 0.7 | 401 | 0.05 |
| walker2d-full-replay | 10.0 | 0.7 | 0.5 | 1.0 | 401 | 0.55 | 0.7 | 401 | 0.1 |
| antmaze-umaze | 0.7 | 10.0 | 0.5 | 3.0 | 101 | 0.65 | 0.9 | 201 | 0.05 |
| antmaze-umaze-diverse | 0.9 | 10.0 | 0.9 | 6.0 | 51 | 0.75 | 0.5 | 201 | 0.05 |
| antmaze-medium-play | 0.9 | 6.0 | 0.9 | 1.0 | 51 | 0.75 | 0.9 | 401 | 0.1 |
| antmaze-medium-diverse | 0.9 | 6.0 | 0.95 | 6.0 | 51 | 0.75 | 0.9 | 401 | 0.1 |
| antmaze-large-play | 0.9 | 10.0 | 0.9 | 10.0 | 101 | 0.75 | 0.9 | 401 | 0.1 |
| antmaze-large-diverse | 0.9 | 6.0 | 0.95 | 3.0 | 201 | 0.55 | 0.9 | 201 | 0.05 |
| pen-human | 0.7 | 0.5 | 0.7 | 1.0 | 201 | 0.85 | 0.9 | 201 | -0.05 |
| pen-cloned | 0.9 | 10.0 | 0.5 | 0.5 | 401 | 0.65 | 0.5 | 401 | 0.1 |
| pen-expert | 0.9 | 0.5 | 0.5 | 10.0 | 401 | 0.75 | 0.5 | 401 | -0.05 |
| door-human | 0.95 | 1.0 | 0.9 | 6.0 | 401 | 0.75 | 0.9 | 201 | 0.1 |
| door-cloned | 0.7 | 1.0 | 0.9 | 0.5 | 21 | 0.75 | 0.7 | 201 | 0.1 |
| door-expert | 0.9 | 10.0 | 0.9 | 6.0 | 101 | 0.55 | 0.7 | 201 | 0.05 |
| hammer-human | 0.9 | 10.0 | 0.7 | 3.0 | 21 | 0.55 | 0.7 | 401 | -0.05 |
| hammer-cloned | 0.7 | 6.0 | 0.7 | 10.0 | 21 | 0.65 | 0.7 | 201 | 0.1 |
| hammer-expert | 0.95 | 0.5 | 0.95 | 0.5 | 401 | 0.65 | 0.9 | 401 | -0.05 |
| relocate-human | 0.7 | 10.0 | 0.7 | 1.0 | 101 | 0.75 | 0.9 | 401 | 0.05 |
| relocate-cloned | 0.7 | 1.0 | 0.7 | 1.0 | 201 | 0.65 | 0.9 | 401 | 0.05 |
| relocate-expert | 0.5 | 0.5 | 0.95 | 0.5 | 401 | 0.75 | 0.9 | 401 | 0.1 |

## C.3  LB-SAC

Table 9: LB-SAC's general hyperparameters.

| Parameter | Value |
|---|---|
| optimizer | Adam Kingma & Ba (2014) |
| batch size | 1024 |
| learning rate (all networks) | 6e-4 |
| tau ($\tau$) | 5e-3 |
| hidden dim (all networks) | 256 |
| num hidden layers (all networks) | 3 |
| gamma ($\gamma$) | 0.99 |
| nonlinearity | ReLU |

Table 10: LB-SAC's best hyperparameters.

| Task Name | N critics (MSE) | N critics (CE+AT) | $m$ (CE+CT) | $\sigma/\zeta$ (CE+CT) | N critics (CE+MT) | $m$ (CE+MT) | $v_{expand}$ (CE+MT) |
|---|---|---|---|---|---|---|---|
| halfcheetah-random | 2 | 50 | 51 | 0.75 | 2 | 201 | 0.1 |
| halfcheetah-medium | 2 | 5 | 101 | 0.75 | 10 | 201 | -0.05 |
| halfcheetah-expert | 5 | 5 | 51 | 0.85 | 10 | 201 | 0.05 |
| halfcheetah-medium-expert | 5 | 5 | 201 | 0.85 | 10 | 201 | -0.05 |
| halfcheetah-medium-replay | 5 | 5 | 201 | 0.85 | 10 | 201 | 0.1 |
| halfcheetah-full-replay | 2 | 2 | 201 | 0.75 | 2 | 401 | -0.05 |
| hopper-random | 5 | 25 | 201 | 0.85 | 25 | 401 | -0.05 |
| hopper-medium | 25 | 25 | 201 | 0.75 | 25 | 401 | 0.1 |
| hopper-expert | 50 | 50 | 201 | 0.65 | 25 | 401 | 0.05 |
| hopper-medium-expert | 50 | 50 | 51 | 0.75 | 25 | 201 | 0.1 |
| hopper-medium-replay | 5 | 5 | 101 | 0.85 | 10 | 201 | 0.05 |
| hopper-full-replay | 5 | 5 | 51 | 0.75 | 10 | 201 | 0.05 |
| walker2d-random | 50 | 5 | 51 | 0.75 | 10 | 201 | 0.05 |
| walker2d-medium | 10 | 10 | 101 | 0.85 | 10 | 201 | -0.05 |
| walker2d-expert | 25 | 50 | 201 | 0.65 | 25 | 401 | -0.05 |
| walker2d-medium-expert | 10 | 25 | 201 | 0.85 | 10 | 401 | 0.05 |
| walker2d-medium-replay | 5 | 5 | 101 | 0.85 | 10 | 201 | 0.1 |
| walker2d-full-replay | 5 | 5 | 201 | 0.85 | 10 | 201 | 0.1 |
| antmaze-umaze | 5 | 5 | 201 | 0.65 | 2 | 201 | 0.1 |
| antmaze-umaze-diverse | 25 | 2 | 101 | 0.75 | 2 | 401 | 0.05 |
| antmaze-medium-play | 25 | 5 | 101 | 0.75 | 2 | 401 | 0.05 |
| antmaze-medium-diverse | 25 | 5 | 101 | 0.75 | 2 | 401 | 0.05 |
| antmaze-large-play | 25 | 25 | 101 | 0.75 | 25 | 201 | 0.1 |
| antmaze-large-diverse | 25 | 25 | 101 | 0.75 | 25 | 201 | 0.1 |
| pen-human | 5 | 10 | 51 | 0.75 | 10 | 201 | 0.05 |
| pen-cloned | 50 | 50 | 51 | 0.65 | 25 | 201 | -0.05 |
| pen-expert | 25 | 25 | 201 | 0.85 | 25 | 201 | 0.1 |
| door-human | 2 | 50 | 51 | 0.75 | 25 | 201 | 0.05 |
| door-cloned | 50 | 50 | 201 | 0.85 | 25 | 201 | 0.05 |
| door-expert | 50 | 50 | 201 | 0.75 | 25 | 401 | 0.05 |
| hammer-human | 5 | 50 | 101 | 0.75 | 2 | 201 | -0.05 |
| hammer-cloned | 50 | 25 | 51 | 0.85 | 10 | 201 | -0.05 |
| hammer-expert | 25 | 25 | 201 | 0.85 | 10 | 401 | 0.05 |
| relocate-human | 2 | 50 | 101 | 0.65 | 25 | 401 | 0.1 |
| relocate-cloned | 50 | 50 | 101 | 0.85 | 25 | 201 | -0.05 |
| relocate-expert | 50 | 25 | 101 | 0.75 | 25 | 401 | -0.05 |

# D    Computational Costs

Table 11: Computational costs for all experiments. Note, ReBRAC computational costs are taken from Tarasov et al. (2024a). The total amount of compute is approximately 23497 hours (979 days).

| Algorithm | Number of runs | Approximate hours per run |
|---|---|---|
| ReBRAC+MSE, tuning | 2784 | 0.39 |
| ReBRAC+CE+AT, tuning | 2784 | 0.34 |
| ReBRAC+CE+CT, tuning | 2880 | 0.48 |
| ReBRAC+CE+MT, tuning | 2592 | 0.50 |
| IQL+MSE, tuning | 2880 | 0.3 |
| IQL+CE+AT, tuning | 2880 | 0.26 |
| IQL+CE+CT, tuning | 2880 | 0.27 |
| IQL+CE+MT, tuning | 2592 | 0.30 |
| LB-SAC+MSE, tuning | 720 | 1.29 |
| LB-SAC+CE+AT, tuning | 720 | 1.61 |
| LB-SAC+CE+CT, tuning | 1296 | 2.32 |
| LB-SAC+CE+MT, tuning | 2592 | 1.56 |
| ReBRAC+MSE, eval | 360 | 0.36 |
| ReBRAC+CE, eval | 144 | 0.45 |
| ReBRAC+CE+AT, eval | 144 | 0.47 |
| ReBRAC+CE+CT, eval | 144 | 0.47 |
| ReBRAC+CE+MT, eval | 144 | 0.34 |
| IQL+MSE, eval | 144 | 0.35 |
| IQL+CE, eval | 144 | 0.33 |
| IQL+CE+AT, eval | 144 | 0.24 |
| IQL+CE+CT, eval | 144 | 0.34 |
| IQL+CE+MT, eval | 144 | 0.34 |
| LB-SAC+MSE, eval | 144 | 1.25 |
| LB-SAC+CE, eval | 144 | 1.34 |
| LB-SAC+CE+AT, eval | 144 | 1.69 |
| LB-SAC+CE+CT, eval | 144 | 2.21 |
| LB-SAC+CE+MT, eval | 144 | 1.89 |
| ReBRAC+MSE, depth scale | 720 | 0.61 |
| ReBRAC+CE, depth scale | 720 | 0.38 |
| IQL+MSE, depth scale | 720 | 0.35 |
| IQL+CE, depth scale | 720 | 0.38 |
| ReBRAC, $v_{expand}$ | 1440 | 0.45 |
| IQL+CE, $v_{expand}$ | 1440 | 0.34 |
| LB-SAC+CE, $v_{expand}$ | 1440 | 2.53 |
| **Sum (w/o ReBRAC MSE)** | 34032 | 0.69 |

# E   MLPs Scale

Table 12: Dependency of the algorithms performance on the number of additional layers averaged over domains.

| Domain | Algorithm | +0 layers | +1 layer | +2 layers | +3 layers | +4 layers |
|---|---|---|---|---|---|---|
| Gym-MuJoCo | **ReBRAC** | 80.3 | 81.4 | 79.7 | 79.1 | 80.1 |
| | **ReBRAC+CE** | 80.3 | 79.2 | 78.8 | 78.7 | 80.0 |
| | **IQL** | 74.0 | 72.3 | 68.1 | 68.3 | 70.7 |
| | **IQL+CE** | 62.4 | 56.5 | 57.7 | 57.0 | 53.4 |
| AntMaze | **ReBRAC** | 76.4 | 75.8 | 61.2 | 55.8 | 52.2 |
| | **ReBRAC+CE** | 89.2 | 88.2 | 86.4 | 87.8 | 81.0 |
| | **IQL** | 64.5 | 52.4 | 52.0 | 46.2 | 49.1 |
| | **IQL+CE** | 17.0 | 18.6 | 15.0 | 16.8 | 18.2 |
| Adroit | **ReBRAC** | 58.1 | 59.5 | 55.0 | 53.4 | 52.6 |
| | **ReBRAC+CE** | 55.2 | 55.9 | 55.3 | 58.8 | 56.0 |
| | **IQL** | 33.0 | 37.4 | 35.2 | 33.3 | 26.4 |
| | **IQL+CE** | 48.9 | 50.9 | 50.9 | 48.7 | 51.7 |

# F   Impact of $m$ and $\sigma/\zeta$

Table 13: Average performance on different domains for a fixed $m$ value and averaged over $\sigma/\zeta$.

| Domain | Algorithm | 21 | 51 | 101 | 201 | 401 |
|---|---|---|---|---|---|---|
| Gym-MuJoCo | **ReBRAC+CE** | 77.6 | 78.0 | 80.2 | 79.5 | 76.7 |
| | **IQL+CE** | 48.4 | 52.6 | 58.9 | 65.4 | 66.6 |
| | **LB-SAC+CE** | - | 59.0 | 65.9 | 65.2 | - |
| AntMaze | **ReBRAC+CE** | 63.7 | 87.1 | 89.4 | 89.1 | 89.2 |
| | **IQL+CE** | 15.5 | 16.4 | 16.8 | 15.9 | 16.1 |
| | **LB-SAC+CE** | - | 6.0 | 8.2 | 9.3 | - |
| Adroit | **ReBRAC+CE** | 56.2 | 56.9 | 55.5 | 57.9 | 58.0 |
| | **IQL+CE** | 49.3 | 47.5 | 49.8 | 50.3 | 50.9 |
| | **LB-SAC+CE** | - | 19.8 | 22.5 | 23.5 | - |
| Average (w/o LB-SAC) | | 51.7 | 56.4 | 58.4 | 59.6 | 59.5 |
| Average | | - | 47.0 | 49.6 | 50.6 | - |

Table 14: Average performance on different domains for a fixed $\sigma/\zeta$ value and averaged over $m$.

| Domain | Algorithm | 0.55 | 0.65 | 0.75 | 0.85 |
|---|---|---|---|---|---|
| Gym-MuJoCo | **ReBRAC+CE** | 78.7 | 78.4 | 78.4 | 78.0 |
| | **IQL+CE** | 59.2 | 58.1 | 58.3 | 57.8 |
| | **LB-SAC+CE** | - | 59.0 | 65.9 | 65.2 |
| AntMaze | **ReBRAC+CE** | 85.3 | 83.0 | 83.8 | 82.7 |
| | **IQL+CE** | 16.1 | 16.5 | 16.3 | 15.7 |
| | **LB-SAC+CE** | - | 6.0 | 8.2 | 9.3 |
| Adroit | **ReBRAC+CE** | 56.8 | 56.8 | 57.3 | 56.6 |
| | **IQL+CE** | 49.4 | 49.7 | 49.7 | 49.4 |
| | **LB-SAC+CE** | - | 19.8 | 22.5 | 23.5 |
| Average (w/o LB-SAC) | | 57.5 | 57.0 | 57.3 | 56.7 |
| Average | | - | 47.4 | 48.9 | 48.6 |

# G   Impact of $v_{min}$ and $v_{max}$

Table 15: Dependency of the algorithm's performance on the $v_{expand}$ parameter.

| Domain | Algorithm | -0.05 | 0.0 | 0.05 | 0.1 | 0.2 |
|---|---|---|---|---|---|---|
| Gym-MuJoCo | **ReBRAC,** *both* | 81.0 | 79.9 | 80.4 | 79.8 | 80.2 |
| | **ReBRAC,** *min* | 71.2 | 79.9 | 80.6 | 80.7 | 79.0 |
| | **IQL,** *both* | 61.3 | 62.7 | 61.1 | 60.2 | 61.1 |
| | **IQL,** *min* | 61.4 | 62.7 | 58.2 | 57.7 | 56.7 |
| | **LB-SAC,** *both* | 59.9 | 64.0 | 66.1 | 67.5 | 65.6 |
| | **LB-SAC,** *min* | 65.8 | 64.0 | 64.7 | 66.1 | 64.5 |
| AntMaze | **ReBRAC,** *both* | 90.1 | 86.3 | 89.0 | 89.3 | 83.6 |
| | **ReBRAC,** *min* | 90.2 | 86.3 | 87.5 | 89.8 | 86.4 |
| | **IQL,** *both* | 16.7 | 16.6 | 34.8 | 33.0 | 22.8 |
| | **IQL,** *min* | 16.9 | 16.6 | 16.2 | 16.2 | 16.2 |
| | **LB-SAC,** *both* | 3.5 | 8.1 | 6.0 | 6.0 | 4.25 |
| | **LB-SAC,** *min* | 10.3 | 8.1 | 7.5 | 7.2 | 3.5 |
| Adroit | **ReBRAC,** *both* | 56.5 | 54.3 | 55.5 | 59.9 | 55.2 |
| | **ReBRAC,** *min* | 56.7 | 54.3 | 56.6 | 57.7 | 56.8 |
| | **IQL,** *both* | 49.7 | 49.2 | 51.8 | 51.1 | 51.0 |
| | **IQL,** *min* | 49.4 | 49.2 | 50.9 | 50.0 | 47.6 |
| | **LB-SAC,** *both* | 23.5 | 22.3 | 24.5 | 22.8 | 19.1 |
| | **LB-SAC,** *min* | 21.3 | 22.3 | 19.3 | 18.8 | 21.0 |

# H    Mixed Tuning Results

Table 16: Average normalized score over the final evaluation and ten (four for LB-SAC) unseen training seeds on Gym-MuJoCo tasks. **CE+MT** denotes cross-entropy with mixed hyperparameters tuning.

| | ReBRAC | | | IQL | | | LB-SAC | | |
|---|---|---|---|---|---|---|---|---|---|
| Task | MSE | CE | CE+MT | MSE | CE | CE+MT | MSE | CE | CE+MT |
| hc-r | $29.5 \pm 1.5$ | $13.4 \pm 0.8$ | $9.2 \pm 0.6$ | $18.9 \pm 1.0$ | $1.9 \pm 0.0$ | $9.0 \pm 3.0$ | $28.2 \pm 1.4$ | $10.0 \pm 0.3$ | $9.9 \pm 0.2$ |
| hc-m | $65.6 \pm 1.0$ | $59.5 \pm 0.7$ | $62.0 \pm 0.5$ | $49.5 \pm 1.1$ | $42.5 \pm 0.3$ | $46.8 \pm 0.1$ | $64.5 \pm 1.3$ | $56.7 \pm 2.1$ | $65.5 \pm 0.5$ |
| hc-e | $105.9 \pm 1.7$ | $103.2 \pm 5.5$ | $104.5 \pm 3.1$ | $95.8 \pm 2.2$ | $92.9 \pm 0.2$ | $94.4 \pm 0.3$ | $103.0 \pm 1.5$ | $103.9 \pm 1.0$ | $98.2 \pm 1.6$ |
| hc-me | $101.1 \pm 5.2$ | $103.5 \pm 4.3$ | $104.4 \pm 2.7$ | $92.3 \pm 2.4$ | $86.5 \pm 4.0$ | $90.7 \pm 3.6$ | $104.5 \pm 2.4$ | $105.4 \pm 2.0$ | $103.2 \pm 2.0$ |
| hc-mr | $51.0 \pm 0.8$ | $50.7 \pm 0.7$ | $52.1 \pm 5.6$ | $45.2 \pm 0.5$ | $38.1 \pm 1.8$ | $43.3 \pm 0.2$ | $52.8 \pm 0.7$ | $55.4 \pm 0.9$ | $54.5 \pm 1.1$ |
| hc-fr | $82.1 \pm 1.1$ | $83.2 \pm 1.5$ | $82.2 \pm 1.7$ | $75.5 \pm 0.5$ | $62.6 \pm 1.1$ | $73.2 \pm 0.5$ | $79.0 \pm 2.0$ | $80.7 \pm 0.3$ | $76.6 \pm 9.7$ |
| hp-r | $8.1 \pm 2.4$ | $8.1 \pm 0.9$ | $8.5 \pm 4.0$ | $6.0 \pm 2.8$ | $14.2 \pm 2.8$ | $9.5 \pm 0.5$ | $14.5 \pm 11.5$ | $8.2 \pm 2.2$ | $14.8 \pm 11.0$ |
| hp-m | $102.0 \pm 1.0$ | $98.9 \pm 9.4$ | $102.0 \pm 0.4$ | $54.8 \pm 4.4$ | $54.1 \pm 2.0$ | $50.1 \pm 5.8$ | $90.0 \pm 27.5$ | $7.9 \pm 0.8$ | $15.6 \pm 13.5$ |
| hp-e | $100.1 \pm 8.3$ | $107.9 \pm 4.9$ | $110.5 \pm 0.4$ | $109.4 \pm 1.8$ | $110.5 \pm 0.5$ | $110.7 \pm 0.2$ | $1.3 \pm 0.0$ | $21.6 \pm 39.7$ | $12.8 \pm 12.8$ |
| hp-me | $107.0 \pm 6.4$ | $111.6 \pm 0.5$ | $105.4 \pm 8.4$ | $88.5 \pm 16.5$ | $64.0 \pm 10.2$ | $79.4 \pm 29.0$ | $111.3 \pm 0.3$ | $14.9 \pm 7.0$ | $9.1 \pm 5.5$ |
| hp-mr | $98.1 \pm 5.3$ | $98.7 \pm 3.0$ | $100.2 \pm 6.4$ | $95.9 \pm 6.4$ | $71.5 \pm 10.1$ | $83.7 \pm 11.2$ | $63.0 \pm 48.1$ | $66.9 \pm 42.8$ | $66.7 \pm 32.7$ |
| hp-fr | $107.1 \pm 0.4$ | $108.2 \pm 0.3$ | $108.5 \pm 0.4$ | $107.2 \pm 0.6$ | $41.9 \pm 4.8$ | $105.7 \pm 0.3$ | $107.0 \pm 0.7$ | $65.6 \pm 50.2$ | $100.3 \pm 2.6$ |
| wl-r | $18.4 \pm 4.5$ | $8.8 \pm 4.4$ | $7.1 \pm 5.4$ | $6.4 \pm 6.3$ | $3.9 \pm 2.1$ | $9.3 \pm 5.7$ | $21.7 \pm 0.0$ | $20.1 \pm 2.9$ | $21.7 \pm 0.1$ |
| wl-m | $82.5 \pm 3.6$ | $85.1 \pm 2.7$ | $84.5 \pm 2.6$ | $83.2 \pm 1.2$ | $79.9 \pm 1.3$ | $80.6 \pm 4.2$ | $89.3 \pm 5.3$ | $89.6 \pm 10.7$ | $95.9 \pm 3.3$ |
| wl-e | $112.3 \pm 0.2$ | $112.7 \pm 0.1$ | $112.3 \pm 0.3$ | $113.8 \pm 0.2$ | $108.5 \pm 0.2$ | $109.5 \pm 0.2$ | $114.2 \pm 0.4$ | $59.8 \pm 45.0$ | $109.8 \pm 1.2$ |
| wl-me | $111.6 \pm 0.3$ | $111.7 \pm 0.1$ | $110.1 \pm 0.2$ | $112.3 \pm 0.6$ | $92.2 \pm 6.5$ | $110.9 \pm 0.3$ | $110.6 \pm 0.4$ | $73.1 \pm 18.1$ | $111.3 \pm 2.5$ |
| wl-mr | $77.3 \pm 7.9$ | $85.2 \pm 6.7$ | $83.3 \pm 14.7$ | $82.0 \pm 7.9$ | $59.1 \pm 8.8$ | $83.3 \pm 3.0$ | $92.6 \pm 2.7$ | $90.9 \pm 5.3$ | $84.3 \pm 8.9$ |
| wl-fr | $102.2 \pm 1.7$ | $101.8 \pm 5.6$ | $100.8 \pm 19.5$ | $97.7 \pm 1.4$ | $85.3 \pm 3.3$ | $93.7 \pm 1.1$ | $102.1 \pm 1.0$ | $110.4 \pm 1.9$ | $95.7 \pm 4.0$ |
| Avg | 81.2 | 80.6 | 80.4 | 74.1 | 61.6 | 71.3 | 74.9 | 57.8 | 63.6 |

Table 17: Average normalized score over the final evaluation and ten (four for LB-SAC) unseen training seeds on AntMaze tasks.

| | ReBRAC | | | IQL | | | LB-SAC | | |
|---|---|---|---|---|---|---|---|---|---|
| Task | MSE | CE | CE+MT | MSE | CE | CE+MT | MSE | CE | CE+MT |
| um | $97.8 \pm 1.0$ | $98.0 \pm 2.1$ | $97.5 \pm 2.0$ | $79.1 \pm 6.8$ | $50.0 \pm 4.6$ | $80.0 \pm 5.2$ | $18.25 \pm 35.8$ | $41.0 \pm 33.2$ | $61.0 \pm 13.7$ |
| um-d | $88.3 \pm 13.0$ | $93.0 \pm 4.5$ | $89.0 \pm 6.3$ | $72.5 \pm 4.5$ | $48.7 \pm 6.9$ | $45.8 \pm 5.5$ | $0.0 \pm 0.0$ | $0.0 \pm 0.0$ | $1.0 \pm 2.0$ |
| med-p | $84.0 \pm 4.2$ | $88.0 \pm 6.3$ | $89.7 \pm 5.0$ | $73.8 \pm 5.4$ | $0.2 \pm 0.4$ | $51.4 \pm 10.0$ | $0.0 \pm 0.0$ | $0.0 \pm 0.0$ | $0.0 \pm 0.0$ |
| med-d | $76.3 \pm 13.5$ | $84.8 \pm 9.3$ | $89.8 \pm 5.4$ | $74.8 \pm 3.8$ | $0.4 \pm 0.9$ | $43.9 \pm 7.1$ | $0.0 \pm 0.0$ | $0.0 \pm 0.0$ | $8.0 \pm 16.0$ |
| lrg-p | $60.4 \pm 26.1$ | $85.9 \pm 6.3$ | $86.2 \pm 6.6$ | $41.3 \pm 7.0$ | $0.0 \pm 0.0$ | $11.8 \pm 1.9$ | $0.0 \pm 0.0$ | $0.0 \pm 0.0$ | $0.0 \pm 0.0$ |
| lrg-d | $54.4 \pm 25.1$ | $87.1 \pm 4.1$ | $80.9 \pm 6.8$ | $23.7 \pm 5.6$ | $0.0 \pm 0.0$ | $4.3 \pm 3.1$ | $0.0 \pm 0.0$ | $0.0 \pm 0.0$ | $0.0 \pm 0.0$ |
| Avg | 76.8 | 89.4 | 88.8 | 60.8 | 16.5 | 39.5 | 3.0 | 6.8 | 11.6 |

Table 18: Average normalized score over the final evaluation and ten (four for LB-SAC) unseen training seeds on Adroit tasks.

| | ReBRAC | | | IQL | | | LB-SAC | | |
|---|---|---|---|---|---|---|---|---|---|
| Task | MSE | CE | CE+MT | MSE | CE | CE+MT | MSE | CE | CE+MT |
| pen-h | 103.5 ± 14.1 | 102.3 ± 10.2 | 86.0± 14.5 | 21.1 ± 16.8 | 108.2 ± 8.4 | 105.0 ± 13.3 | 4.5 ± 2.6 | 7.1 ± 5.6 | 6.2 ± 8.5 |
| pen-c | 91.8 ± 21.7 | 90.3 ± 14.2 | 102.2 ± 20.4 | 13.2 ± 21.6 | 12.5 ± 14.7 | 100.8 ± 11.9 | 26.1 ± 5.3 | 20.0 ± 4.5 | 16.6 ± 8.5 |
| pen-e | 154.1 ± 5.4 | 155.0 ± 5.8 | 157.9 ± 1.2 | 60.2 ± 37.7 | 134.9 ± 7.0 | 147.0 ± 6.6 | 130.5 ± 16.8 | 38.0 ± 13.2 | 55.3 ± 50.6 |
| door-h | 0.0 ± 0.0 | 0.0 ± 0.0 | 0.0 ± 0.0 | 5.9 ± 2.5 | 4.3 ± 1.0 | 0.8 ± 0.5 | -0.2 ± 0.1 | -0.2 ± 0.1 | -0.2 ± 0.1 |
| door-c | 1.1 ± 2.6 | 0.0 ± 0.0 | 0.0 ± 0.0 | 0.2 ± 0.3 | 0.3 ± 0.3 | 6.3 ± 1.8 | 0.0 ± 0.0 | 0.2 ± 0.5 | 0.0 ± 0.0 |
| door-e | 104.6 ± 2.4 | 105.7 ± 1.5 | 105.7 ± 1.4 | 105.4 ± 2.0 | 106.1 ± 0.5 | 105.8 ± 0.0 | 95.0 ± 8.6 | 70.6 ± 33.8 | 80.6 ± 6.2 |
| ham-h | 0.2 ± 0.2 | 0.1 ± 0.1 | 2.1 ± 2.7 | 1.7 ± 0.8 | 1.5 ± 0.7 | 3.8 ± 5.2 | 0.1 ± 0.0 | 0.0 ± 0.0 | 0.1 ± 0.0 |
| ham-c | 6.7 ± 3.7 | 9.7 ± 10.4 | 2.9 ± 2.4 | 0.2 ± 0.0 | 1.0 ± 0.7 | 0.9 ± 0.6 | 20.2 ± 16.8 | 13.6 ± 15.4 | 0.3 ± 0.5 |
| ham-e | 133.8 ± 0.7 | 122.9 ± 19.7 | 130.5 ± 11.9 | 129.5 ± 0.2 | 129.5 ± 0.1 | 128.8 ± 0.1 | 76.6 ± 59.5 | 91.1 ± 10.5 | 129.3 ± 14.7 |
| rel-h | 0.0 ± 0.0 | 0.0 ± 0.0 | 0.0 ± 0.2 | 0.1 ± 0.1 | 0.1 ± 0.0 | 0.1 ± 0.0 | 0.0 ± 0.0 | -0.1 ± 0.0 | 0.0 ± 0.0 |
| rel-c | 0.9 ± 1.6 | 0.4 ± 0.5 | 0.2 ± 0.2 | 0.1 ± 0.1 | 0.1 ± 0.1 | 0.1 ± 0.1 | 0.0 ± 0.0 | -0.1 ± 0.0 | 0.0 ± 0.0 |
| rel-e | 106.6 ± 3.2 | 107.8 ± 3.3 | 108.5 ± 2.2 | 108.2 ± 0.9 | 105.7 ± 1.8 | 110.0 ± 0.7 | 26.7 ± 18.8 | 5.3 ± 3.5 | 0.6 ± 0.2 |
| Avg w/o e | 25.5 | 25.3 | 24.1 | 5.3 | 16.0 | 27.2 | 6.3 | 5.0 | 2.8 |
| Avg | 58.6 | 57.8 | 58.0 | 37.1 | 50.3 | 59.1 | 31.6 | 20.4 | 24.0 |

# I   rliable Metrics

## I.1   All domains

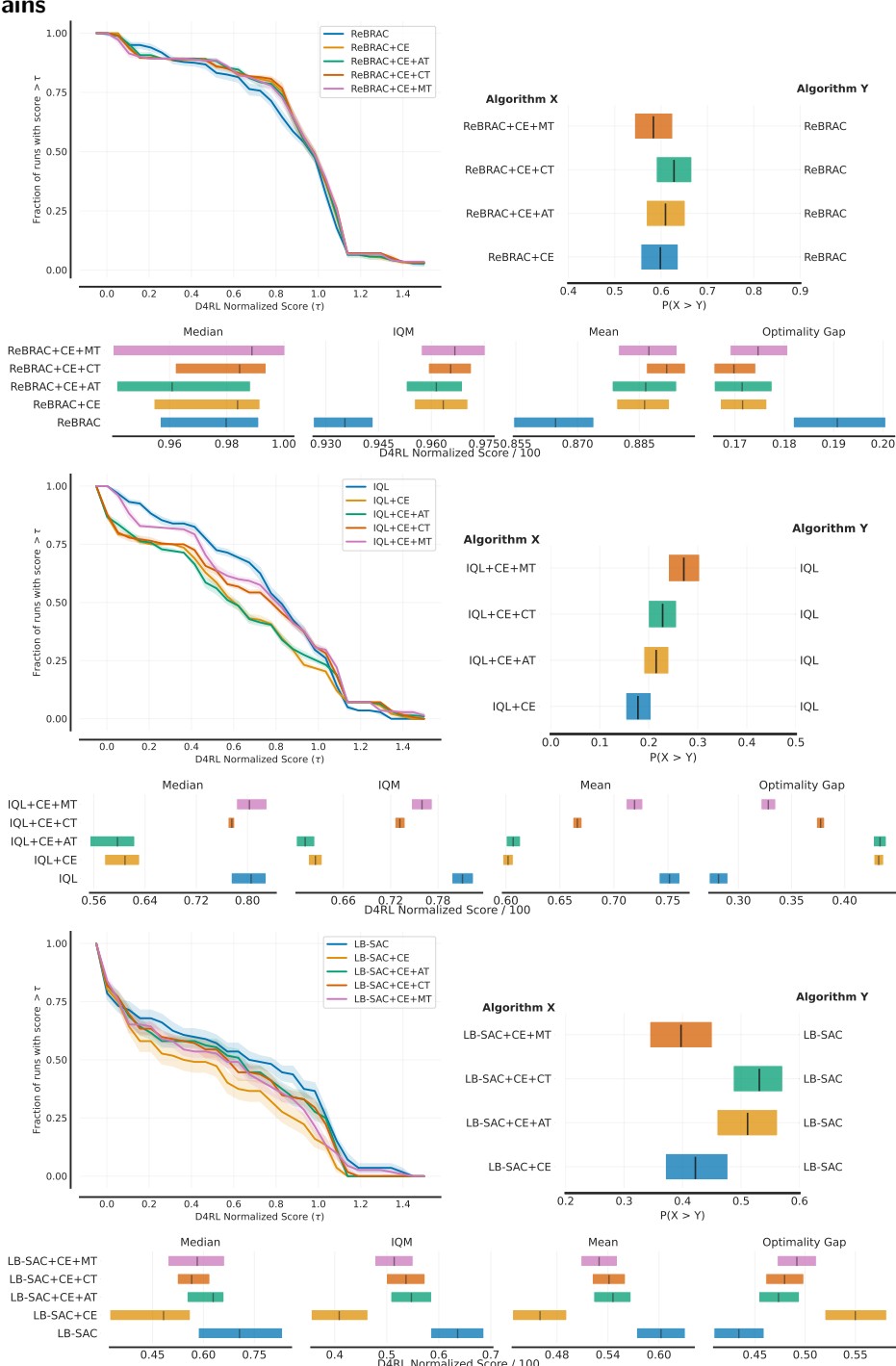

Figure 5: rliable (Agarwal et al., 2021) metrics for ReBRAC, IQL, and LB-SAC averaged over all Gym-MuJoCo, AntMaze and Adroit datasets. Ten evaluation seeds are used for ReBRAC and IQL and four seeds for LB-SAC.

## I.2    Gym-MuJoCo

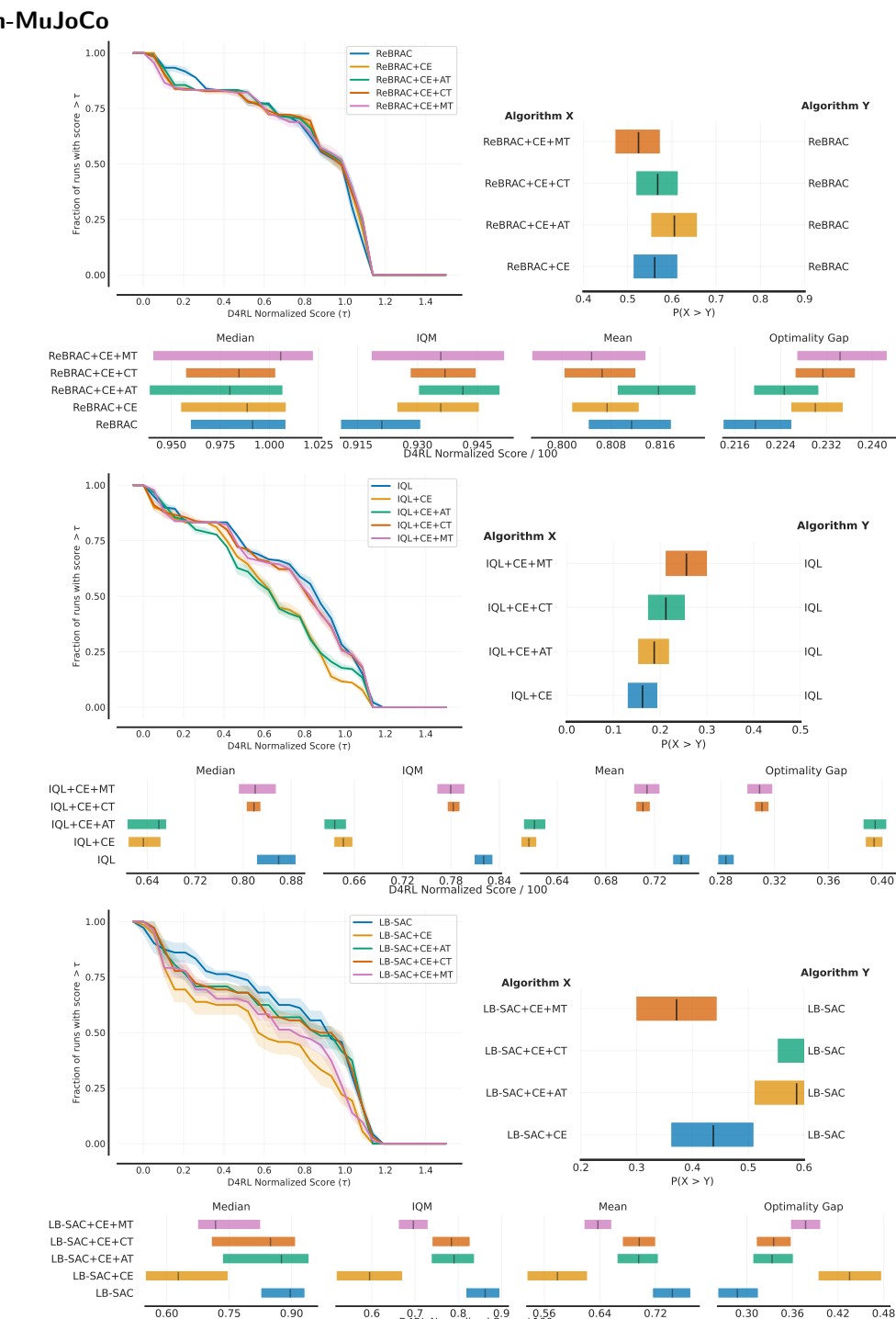

Figure 6: rliable (Agarwal et al., 2021) metrics for ReBRAC, IQL, and LB-SAC averaged over Gym-MuJoCo datasets. Ten evaluation seeds are used for ReBRAC and IQL and four seeds for LB-SAC.

## I.3   AntMaze

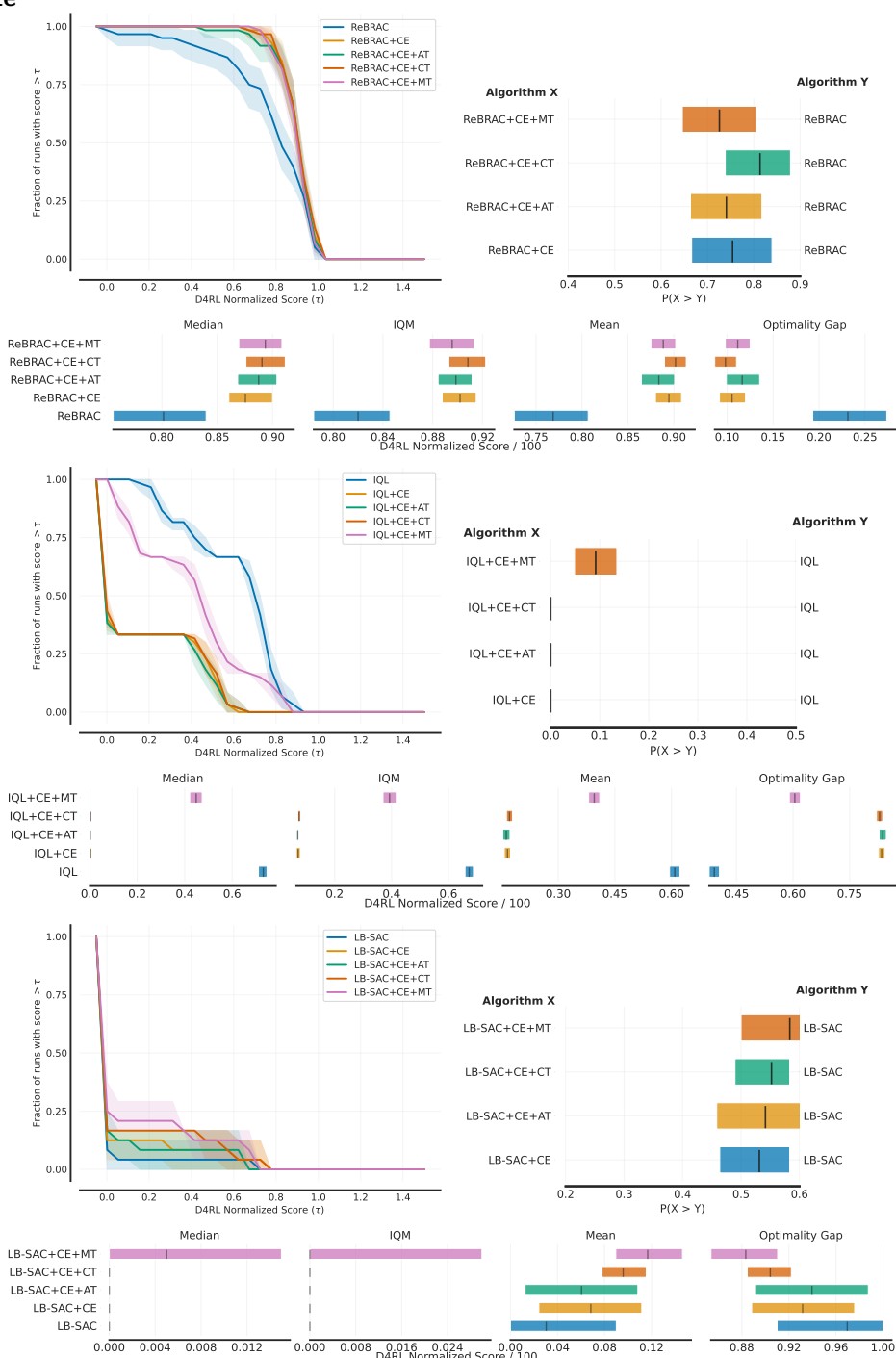

Figure 7: rliable (Agarwal et al., 2021) metrics for ReBRAC, IQL, and LB-SAC averaged over AntMaze datasets. Ten evaluation seeds are used for ReBRAC and IQL and four seeds for LB-SAC.

## I.4 Adroit

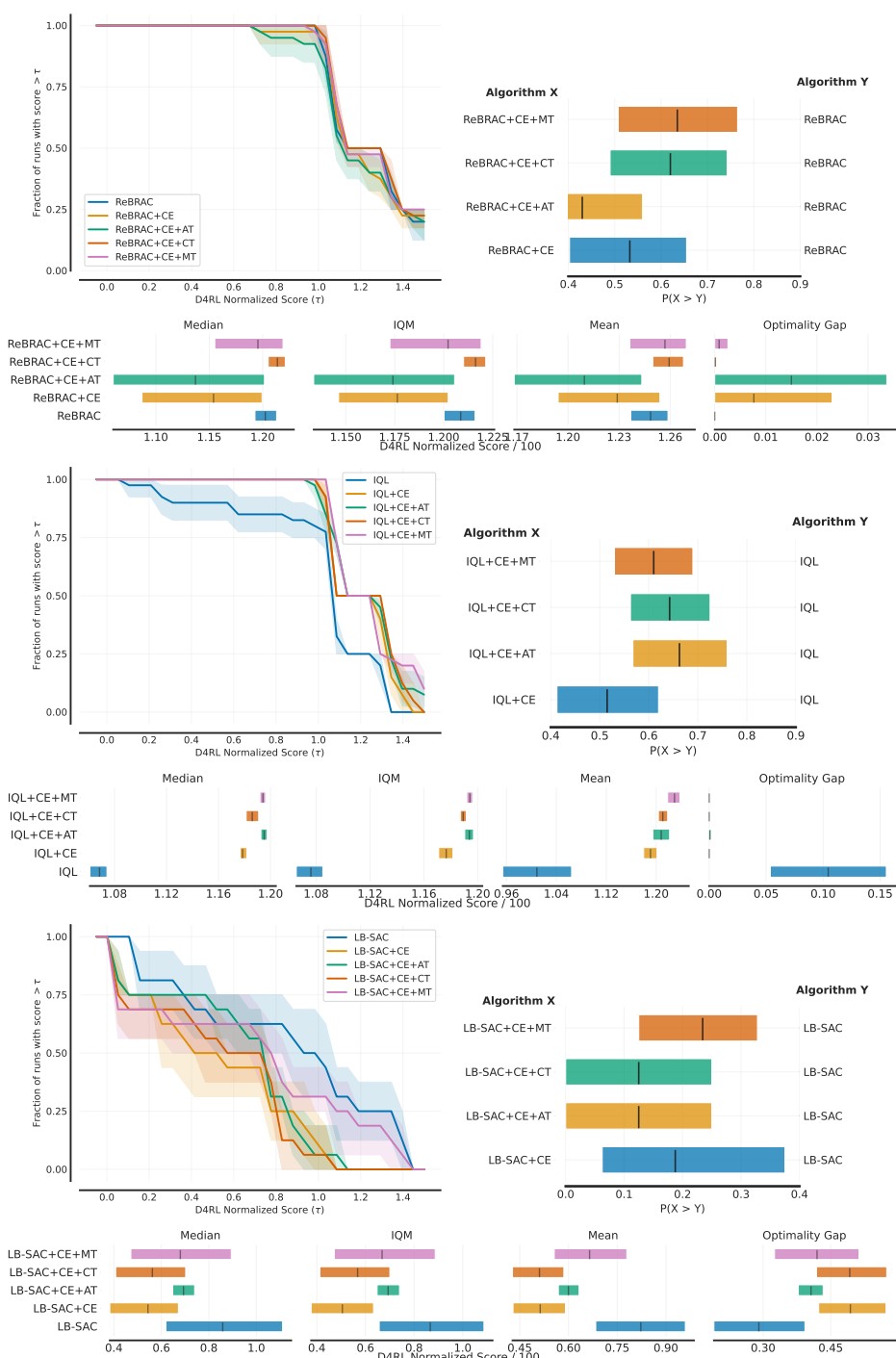

Figure 8: rliable (Agarwal et al., 2021) metrics for ReBRAC, IQL, and LB-SAC averaged over Adroit datasets. Ten evaluation seeds are used for ReBRAC and IQL and four seeds for LB-SAC.

## J Classification Parameters Performance Heatmaps

### J.1 ReBRAC, Gym-MuJoCo

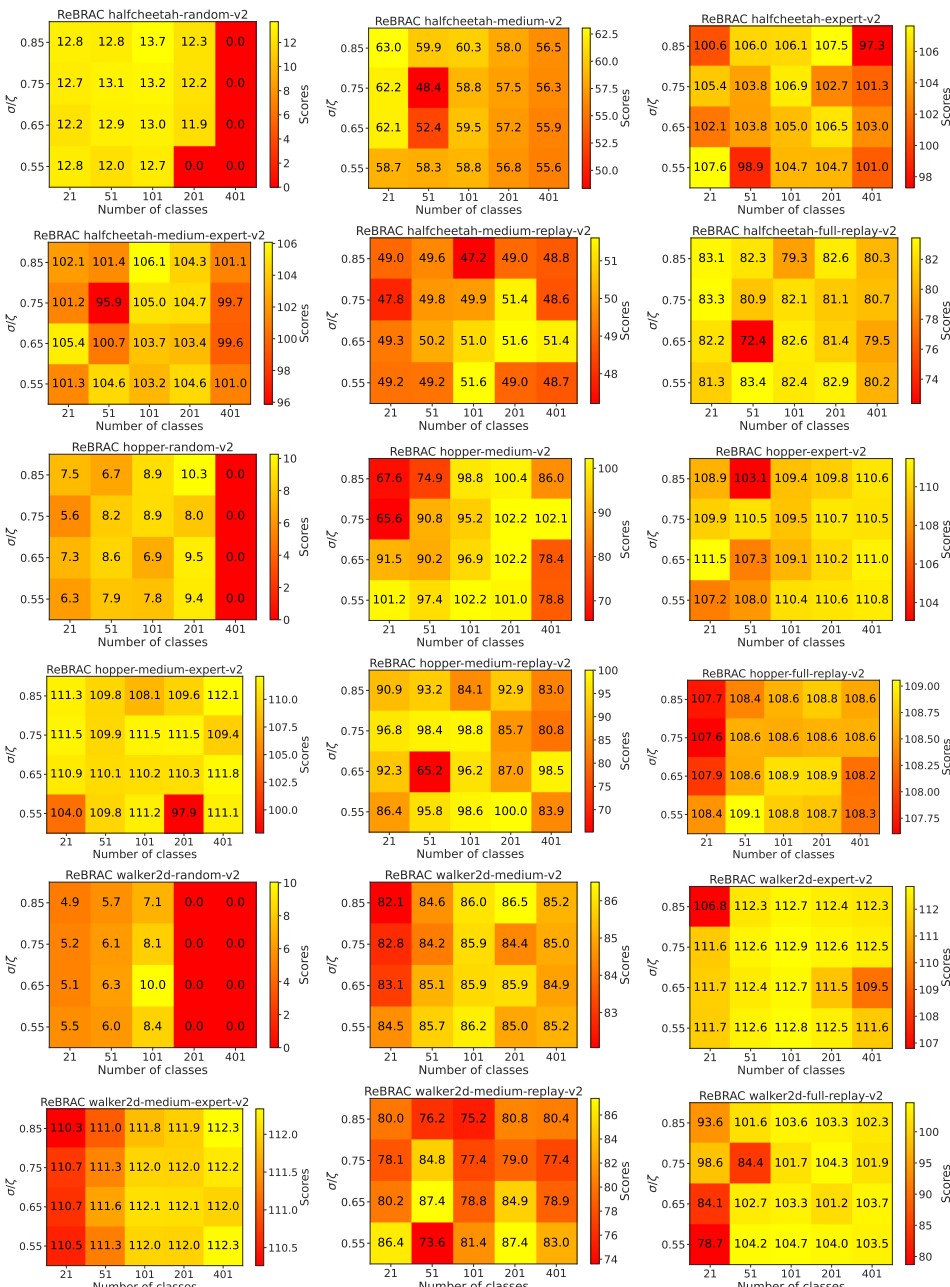

Figure 9: Heatmaps for the impact of the classification parameters on Gym-MuJoCo datasets for ReBRAC.

## J.2 ReBRAC, AntMaze

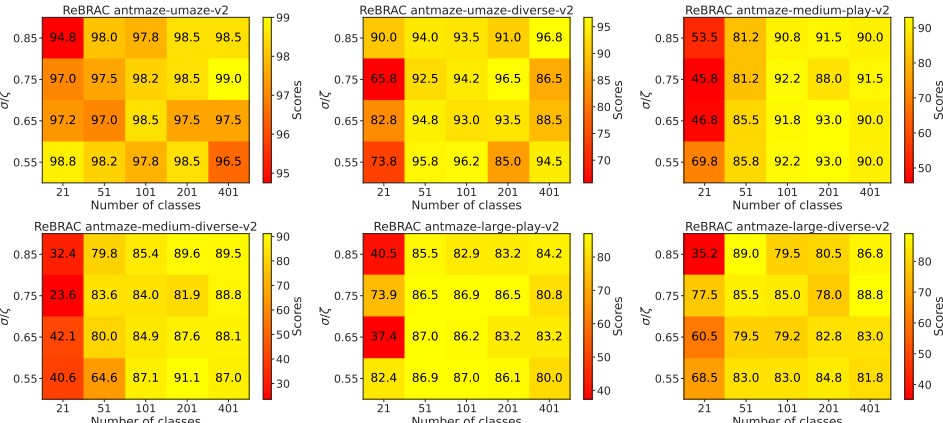

Figure 10: Heatmaps for the impact of the classification parameters on AntMaze datasets for ReBRAC.

## J.3 ReBRAC, Adroit

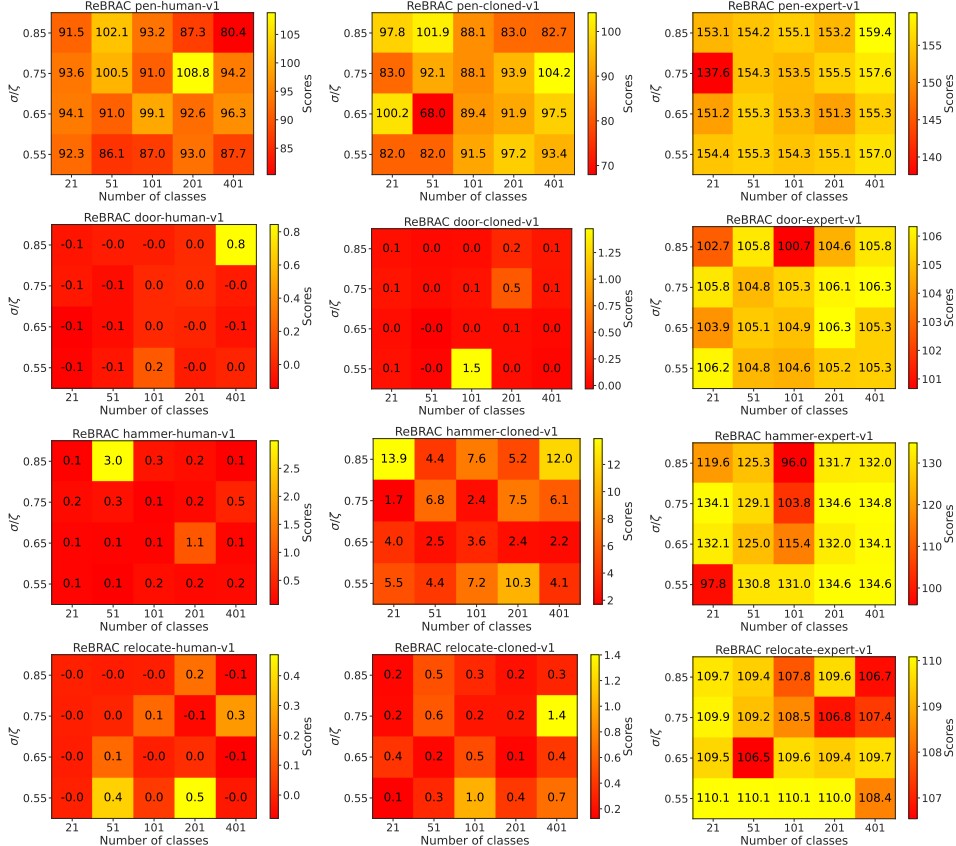

Figure 11: Heatmaps for the impact of the classification parameters on Adroit datasets for ReBRAC.

## J.4    IQL, Gym-MuJoCo

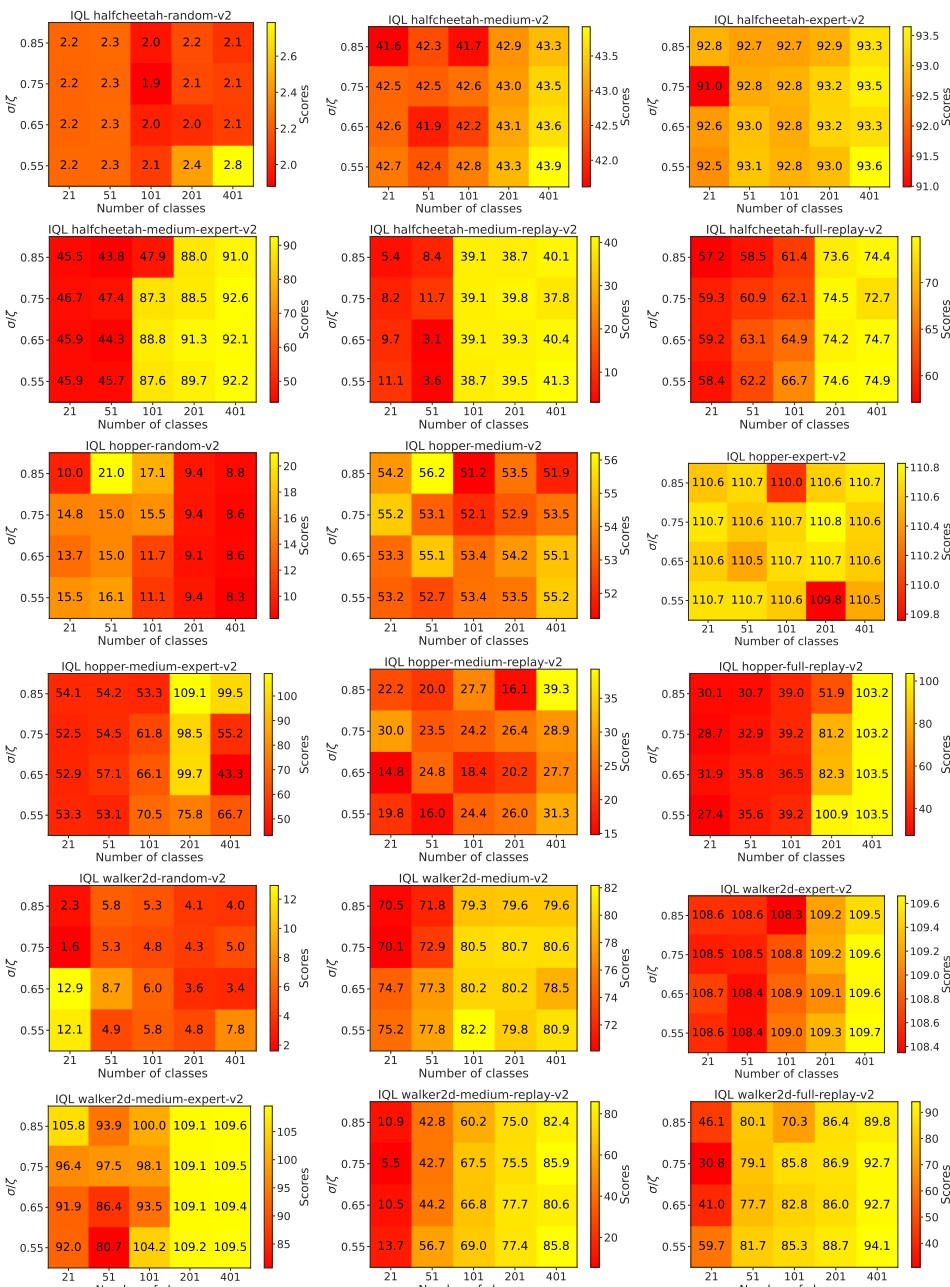

Figure 12: Heatmaps for the impact of the classification parameters on Gym-MuJoCo datasets for IQL.

## J.5 IQL, AntMaze

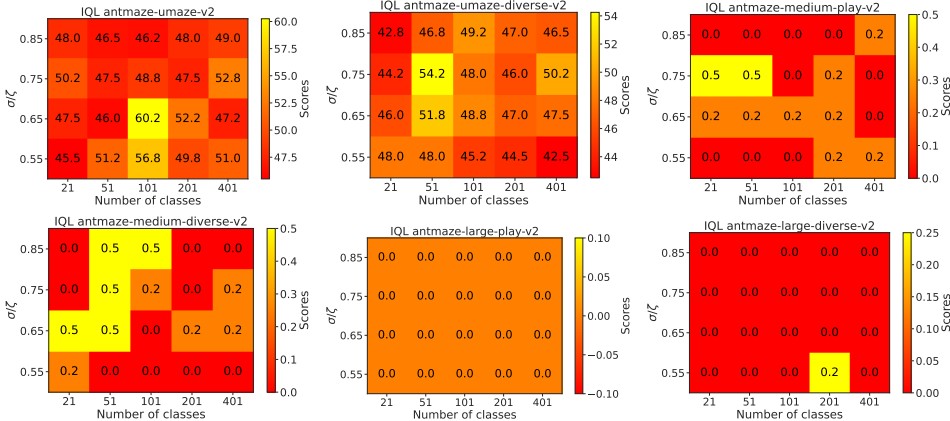

Figure 13: Heatmaps for the impact of the classification parameters on AntMaze datasets for IQL.

## J.6 IQL, Adroit

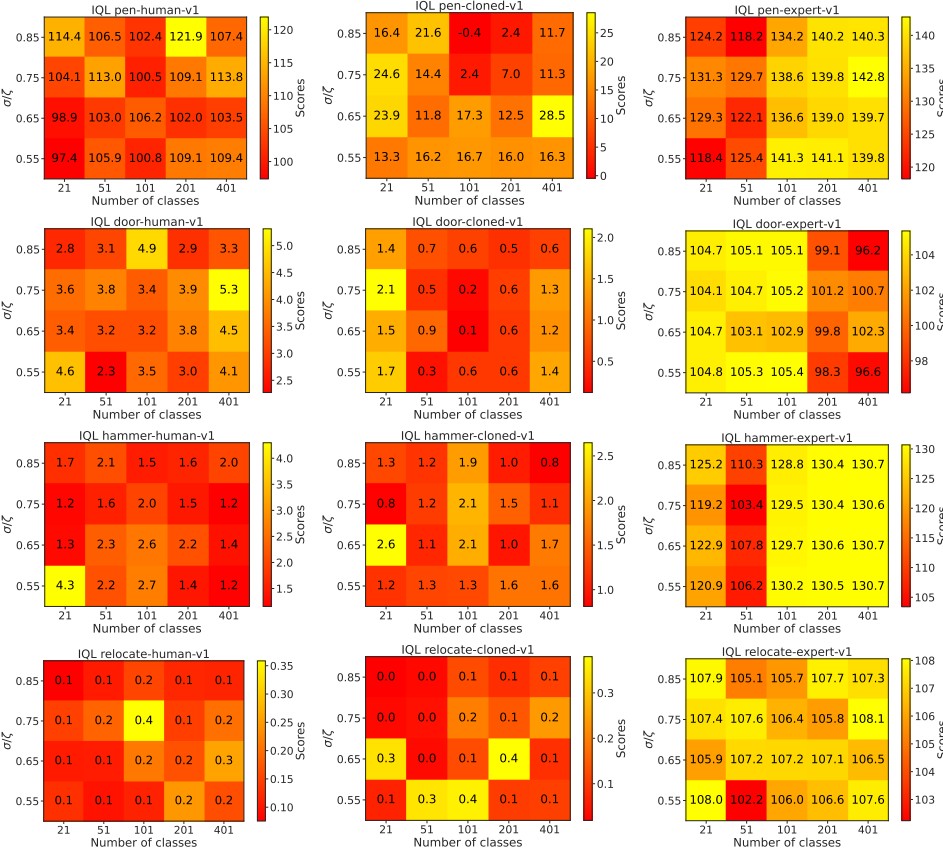

Figure 14: Heatmaps for the impact of the classification parameters on Adroit datasets for IQL.

## J.7 LB-SAC, Gym-MuJoCo

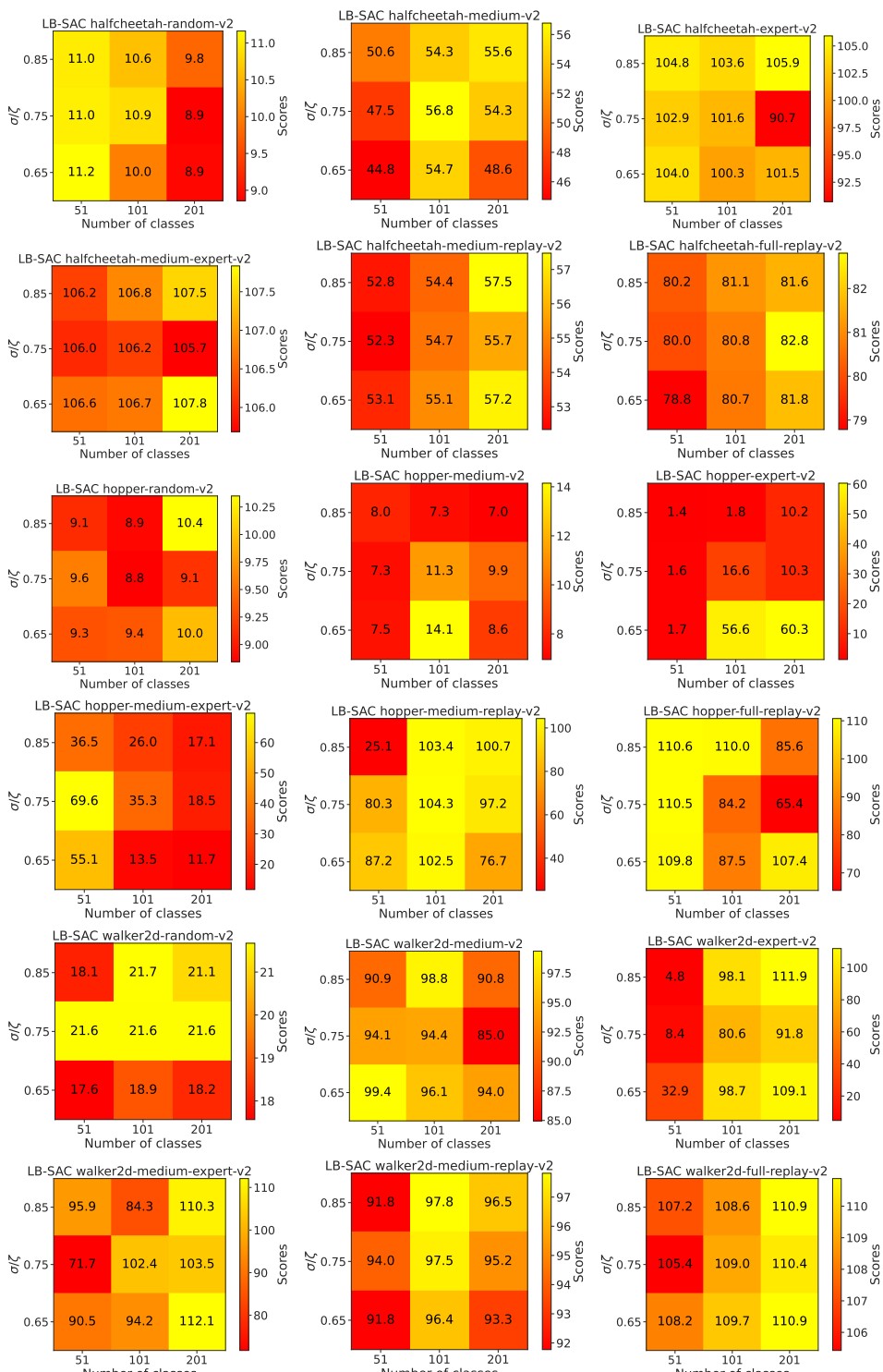

Figure 15: Heatmaps for the impact of the classification parameters on Gym-MuJoCo datasets for LB-SAC.

## J.8 LB-SAC, AntMaze

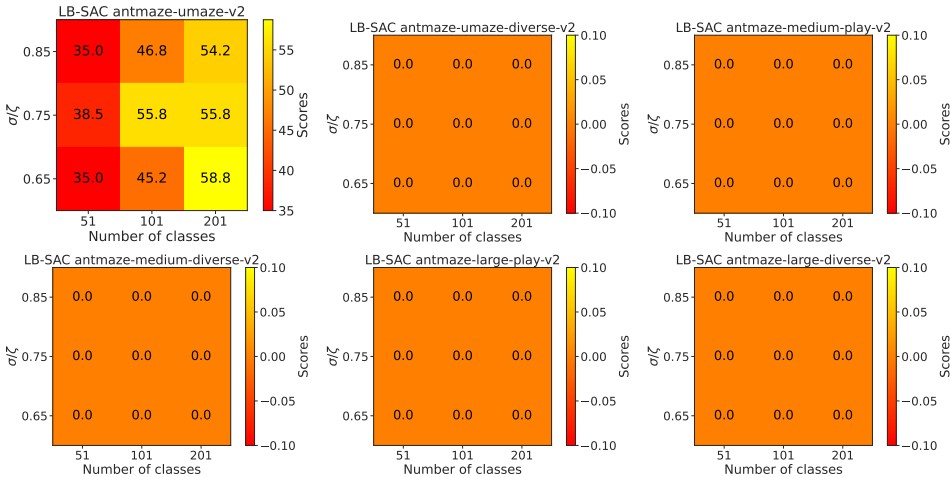

Figure 16: Heatmaps for the impact of the classification parameters on AntMaze datasets for LB-SAC.

### J.9    LB-SAC, Adroit

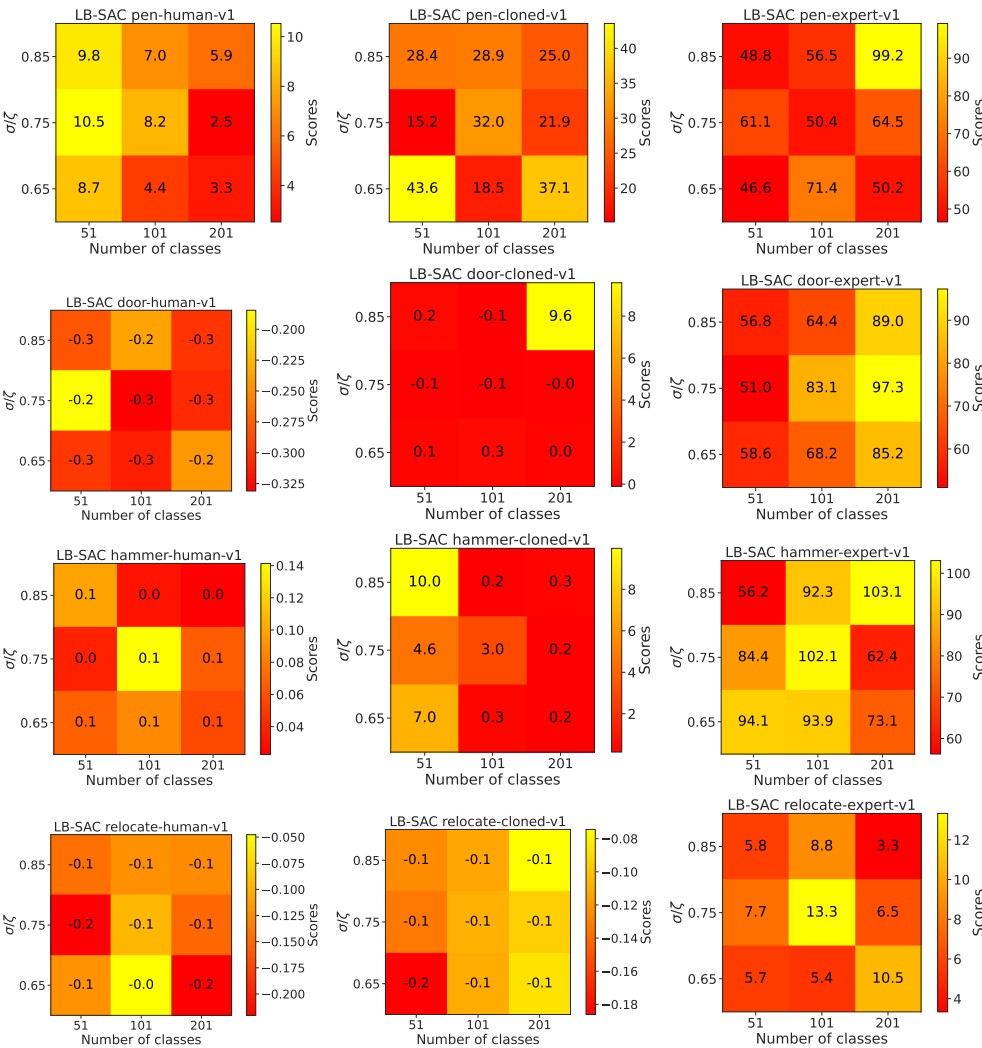

Figure 17: Heatmaps for the impact of the classification parameters on Adroit datasets for LB-SAC.

