# OpenReview forum: "Is Value Functions Estimation with Classification Plug-and- play for Offline Reinforcement Learning?"
_TMLR — Accepted by TMLR_

### Review · Reviewer_MQhN · 2024-09-28

**Summary Of Contributions:**

The paper studies the effects of using the cross-entropy loss for value (q-functions) learning in offline RL. For this, it modifies 3 existing, well-known value-based offline RL approaches (ReBRAC, IQL, and LB-SAC) to use this loss for the critic update and systematically evaluates the effects on the d4rl benchmarks (specifically, the standard gym tasks, ant maze, and the adroit hand tasks). Furthermore, it analyses how using the cross-entropy loss affects both the optimal hyperparameters, as well as finding those in the first place.

**Audience:**

Yes

**Broader Impact Concerns:**

n.a.

**Claims And Evidence:**

No

**Requested Changes:**

- Clarify / Add metrics that allow assessing the statistical significance for all results and statements (see above)
- Rework readability of plots

*Not required but appreciated:*
- Explicitly answering the 4 questions posed in the introduction in the conclusion, based on the findings of the work.

**Strengths And Weaknesses:**

To me, using cross-entropy for value learning is among the most interesting recent trends in Deep RL and the paper presents a valuable study to evaluate this in the context of offline RL. The experiments elucidate the benefits (or lack thereof) across a range of different algorithms and domains of different hyperparameters, in particular those added by introducing the binning for cross-entropy. I highly appreciate the studies including so-far under-explored aspects (beyond considering offline rl instead of online), such as choosing v_min and v_max,  and the scaling analysis for MLPs used in most of RL.
The paper is overall clearly written, easy to follow, and provides a valuable and well motivated reference to the (offline) RL community.

The paper’s key weakness is the lack of rigor applied to establish statistical significance (or lack thereof) in the results:
- It is a bit unclear to me how many seeds were used. Section 4.1. state 4 seeds were used for final evaluation, but the table captions say ten were used, except for LB-SAC which has only 4. For the final evaluation, I believe 10 should be used for all approaches (also LB-SAC).
- What exactly is indicated by the +- values in the table? Are those standard deviations or standard errors or something else?
- Most of the statements in the evaluation consider the performance of a certain algorithm on a certain set of tasks (gym, ant, or adroit) but no values for statistical uncertainty are provided for the “avg” rows in the tables or the entries in the heatmaps. (There are some rliable metrics in the supplement, but they only consider the algorithms across all 3 task sets and only for the findings in tables 1-3)

Additionally, I found some of the plots and heatmaps hard to read, mostly due to very small font sizes (plots and heatmaps) and poor contrast (heatmaps, white text on yellow tiles, black text on dark blue tiles), and believe this could be easily improved.

As a final suggestion for improvement, I would highly appreciate a “key takeaways” or “best practices” section in the conclusion. The paper provides a lot of experiments and answers all questions stated in the introduction thought setion 4, but I think a concise, explicit recap of the answers to those questions in the conclusion would highly improve the accessibility of the results.

---

> ### Author Response · Authors · 2024-10-05
>
> We thank reviewer for their work. We will upload the updated PDF after resolving all reviewers requests and notify about update after that.
>
> # Addressing questions
>
> 1. We use 10 evaluation seeds for ReBRAC and IQL and 4 for LB-SAC. Thanks for noticing inconsistency in Section 4.1, we will fix it in updated script. Unfortunately, LB-SAC requires too much compute to run and we do not have access to computational resources that we had before anymore, so we are unable to run more evaluations.
> 2. Those are stds across random seeds. We will explicitly add this to captions.
> 3. Would you like to see rliable metrics for each domain individually? We did not see much reason to add it as we are usually interested in effects over various domains. rliable metrics we provide demonstrate that ReBRAC is expected to improve on average, LB-SAC is expected to get worse and for IQL situation is mixed. And this observations are presented with statistical significance -- rliable error bars. We can not come up with a good avg uncertainty score for avg in tables due to the high variance in different tasks scales, so metric like std is completely uninformative there. It is a common practice to report avg performances across different tasks without uncertainty.
>
> # Addressing changes
> 1. We hope that our answer clarifies metrics and changes that we will introduce address reviewer's concerns  regarding this problem. We are hoping for clarification about question (3).
> 2. We increased font sizes for plots and improved contrast for heatmaps. Updated version of the paper will contain these changes.

---

> > ### Comment · Reviewer_MQhN · 2024-10-22
> >
> > Thank you for your clarifications and my apologies for the late response.
> >
> > Regarding 3: I agree that the main takeaways are from observations across all tasks and that the already provided reliable metrics support these. However, I would still appreciate the metrics for the individual domains (which can be in the appendix), as quite a few statements in Sections 4.2 and 4.3 contrast the effects on different domains. It would also remove the issue with the missing uncertainty quantification for the average values (where I agree that naively taking the std for the tables is problematic).

---

> > > ### Author Response · Authors · 2024-10-22
> > >
> > > We added the rliable per-domain metrics in Appendix I. The updated script should be available to you.

---

### Review · Reviewer_aTU5 · 2024-09-29

**Summary Of Contributions:**

This paper studies the effectiveness of integrating classification loss with offline RL problems, in particular, three offline RL algorithms are extensively studied, where the corresponding bellman function error functions are modified by cross-entropy losses.

The authors first show that the integration of the classification loss with offline RL is seamless. Secondly, various tuning schemes are developed as a tool to analyze the performances/sensitivities of the RL algorithms. Empirical results show that by selecting the proper hyper-parameters, solving offline RL problems via classification can outperform classic L2 loss. However, the effectiveness of such modification is not consistent across different offline RL algorithms.

Additionally, the authors show that the performance cannot be improved by simply enlarging the value function neural network (MLP) size.

**Audience:**

No

**Broader Impact Concerns:**

N.A.

**Claims And Evidence:**

Yes

**Requested Changes:**

Sections 2-3 should include more details about the introduction to offline RL training and corresponding parameters. Section 3 should include some arguments on how and why the three algorithms are selected.

**Strengths And Weaknesses:**

**Strengths**

---


**1.** The authors show that the classification loss can be plugged easily into three offline RL algorithms. This is a promising result as it will not involve massive modifications to open-source codes and can be directly used by practitioners. The results shown in this paper should be interesting to many researchers in the offline RL field.


---

**2.** The authors illustrate that the performance improvements of adopting classification loss are algorithm-dependent. Both when the algorithms or the classification parameters are tuned, certain algorithms can improve substantially while some algorithms are not sensitive to hyper-parameters. The observations in this paper can guide future researchers to tune their methods when applying classification loss to other offline RL algorithms. In some sense, a hope of improvement should not be certainly expected by a simple integration.

---


**3.** A previous paper shows that the scalability of value function models can be greatly enhanced when parametrizing the value functions by Transformers or ResNets and using the classification loss. However, this paper gives a negative result on classic multilayer perceptrons.

---

**Weaknesses**

---

**1.** Three specific offline RL algorithms are analyzed, which are used to illustrate that the classification loss is indeed a "plug-and-play" replacement for offline RL algorithms. However, I think the three algorithms are not enough to show that the simple integration always hold. Does there exist some algorithms that cannot be easily "combined"? It will be very helpful and make the whole framework more complete if the authors can comment on general offline RL algorithms or provide some counter-examples. In addition, the authors can discuss the reason for selecting these three algorithms as representatives for other offline algorithms.

---

**2.**  The preliminaries and methodology section are not well-presented. Some notations are used without explanations. See my comments below:

a. On Page 2, "Markov Decision Process characterized by a tuple S, A, P, R, γ,", it would be better to include parentheses.

b. On Page 2, "to determine a policy (agent) π", I think the descriptions of agent and policy are not clear and it might be better to drop the agent.

c. On Pages 2-3, The author can put more details when introducing different TD loss in section 2.2., e.g., the HL-Gauss method and the meaning of the parameters in the HL-Gauss method $\sigma$ and $\zeta$. Besides, the meaning of the ratio and the reason that 0.75 is a good choice. I understand those results have been shown in a previous paper, it will help the reader to better understand this paper if the background introduction can include more details and be more complete.

d. On Pages 3-4,  "policy regularization" and "computational constraints" are used without too many details. The authors could comment on which parts of the algorithms that make the computation complicated.

---

**3.** The authors have stated that the random offline datasets are a common issue when stating if something can help the performance. Have the authors done some experiments on the performance of different algorithms when the offline datasets are different? I believe this could be very helpful as the quality of the offline dataset and the differences between the offline dataset and the testing dataset could play a very significant role in the final performance.

---

> ### Author Response · Authors · 2024-10-05
>
> Many thanks for reviewing our work. We will upload the updated PDF after resolving all reviewers requests and notify about update after that.
>
> # Addressing weaknesses
> 1. Most of the existing offline RL approaches are actor-critic algorithms and belong to one of the following families: policy regularization, implicit regularization or Q-function regularization.  In our study we provide the results for three representative algorithms from each of the family. This makes our study representative enough as most of the other approaches are very similar to those that we consider. We are not aware about any offline RL actor-critic algorithm where the proposed framework is inapplicable.
> 2.  c. HL-Gauss description is provided in the section and we also described all of the parameters meanings. Could you please explain what needs to be changed? $\zeta$ is explained here:
> "The approach involves parameterizing the Q-function as a distribution over returns, segmented into $m$ bins with widths $\zeta$, where the first bin corresponds to the predefined $v_{min}$ value, and the last to $v_{max}$."
> And $\sigma$ here:
> "HL-Gauss employs a normal distribution analog for mapping values into neighboring bins, with a hyperparameter $\sigma$ determining the distribution's breadth."
> 3. Random datasets are part of the D4RL benchmark and reported under "-r" in tables. All of the other D4RL datasets are not random. Probably we misunderstand the issue here, could you please elaborate?
>
> # Addressing changes
> We are improving section 2 and 3 based on the reviewers comments in the new version of the script.

---

> > ### Comment · Reviewer_aTU5 · 2024-10-23
> >
> > Thank you for your clarifications and sorry for the late reply.
> >
> > The authors have addressed most of my concerns and questions in the updated draft (I missed some notations in the first round review).
> >
> >
> > Let me elaborate on the random datasets point: you mentioned that "these hyperparameter adjustments ...., although random datasets remained problematic " at the bottom of page 5 in the new draft. Does the randomness here mean the testing datasets?
> >
> > For general offline RL problems, the algorithm's performance is quite sensitive to the training dataset. In the paper, you use the whole datasets from the D4RL benchmark. Have you tried sampling a minibatch from the whole dataset and training the algorithm? I wonder how stable/sensitive the offline RL with classification algorithms are in terms of the randomness from the training set.

---

> > > ### Author Response · Authors · 2024-10-23
> > >
> > > Random refers to D4RL training datasets with "-random" suffix in names where the data was collected with a random policy. By some reason adding classification obejctive strongly decrease the performance on this type of datasets.
> > >
> > > No, we haven't tried to use the subsets of original datasets due to computational constraints. However, some D4RL datasets are subsets of others, e.g. "-medium" and "-expert" are subsets of "-medium-expert", or "-human" are subsets of "-cloned"

---

### Review · Reviewer_DVpN · 2024-10-01

**Summary Of Contributions:**

Following the recent results of Farebrother et al. (2024) which establish that
training value functions with a classification-based loss enables superior
performance and scaling in RL, this work examines whether the phenomenon holds
across a variety of offline RL methods in continuous control. Through a wealth
of experiments, the authors find that classification losses can be quite
sensitive to their additional hyperparameters in RL, and overall did not exhibit
superior performance with as much consistency as they expected. The authors
thoroughly examine additional hyperparameters to tune for the classification
loss and introduce new parameters that can affect quantization (such as a form
of 'padding' at the return boundaries), and highlight the difficulty of
finding a uniformly-good set of hyperparameters.

**Audience:**

Yes

**Broader Impact Concerns:**

No broader impact concerns.

**Claims And Evidence:**

No

**Requested Changes:**

1.  There is a typo in section 2.1 &#x2014; &ldquo;The objective RL of is to&#x2026;&rdquo; should be
    &ldquo;The objective of RL is to&#x2026;&rdquo;.
2.  There is a typo in the IQL section on page 3 &#x2014; &ldquo;We opt for IQL as it the
    best example&#x2026;&rdquo; is missing a word.
3.  Based on the motivation for using the cross-entropy loss in RL, I would
    expect it to benefit the actor (in the case of continuous action spaces) as
    well. However, your code still uses regression losses for the actor, and I
    think this should be discussed.
4.  There should be some pseudocode, even if only in the appendix, to depict how
    you adapt the ReBRAC method for classification loss. My understanding is you
    compute the target according to the following steps:

    1.  Compute $Q(s', a', \hat\theta)$ by taking the mean of the $Z(s', a', \hat\theta)$;
    2.  Compute $\mathcal{T}Q = R(s, a) + \gamma (Q(s', a', \hat\theta) - \beta(a' - \hat{a'})^2)$;
    3.  Construct a target distribution $\hat{Z} = \mathsf{HL-Gauss}(\mathcal{T}Q)$.

    I skimmed the source code, and this appears to be correct.
    Now, I&rsquo;d expect the action penalty and $\beta$ to highly influence the support of
    the distributions $Z(s, a, \theta)$ &#x2013; how do you account for this?
5.  Same as above for IQL. Also, why is it appropriate to train the $V$ function by
    expectile regression if the goal is to train the action-values by classification?
6.  Same as above for LB-SAC.
7.  In the LB-SAC section (in section 3), the estimate of the minimum $Q$-value
    under a Gaussian distribution ensemble isn&rsquo;t clearly relevant. Does this
    pertain specifically to the classification loss? I suspect not, since the
    same equation is given in the LB-SAC paper, so I would recommend removing
    this here because it&rsquo;s unclear what the takeaway is.
8.  There is a typo in section 4.1 &#x2014; &ldquo;When exploring parameter search in, we
    utilized&#x2026;&rdquo; is missing something before the comma.
9.  Using the same hyperparameter grid for ReBRAC and IQL seems unfair due to the
    concerns I pointed out above. Namely, in ReBRAC, the support of the
    distribution representations should be fundamentally different. Moreover,
    these two algorithms do not have the same hyperparameters, so what does it
    mean to use the same hyperparameter grid for the two?
10. Why are there no CE+AT+CT columns in Tables 1-3, that is, results where the algorithm
    parameters *and* the classification parameters are tuned?

**Strengths And Weaknesses:**

## Strengths

This paper presents an enormous amount of experiments, and it is clear that the
authors dove deep and rigorously evaluated many properties of the algorithms
they tested. In particularly, I commend the authors for their experiments
assessing the influence of $\sigma/\zeta$ as well as their treatment of the distribution
support width, with the introduction of the $v_{expand}$ parameter which was
interesting. I really appreciated the heatmap plots (e.g. Figure 1), which
effectively conveyed the message that it is difficult (and maybe generally
impossible) to find a single set of classification hyperparameters that will
work well across algorithms and domains.


## Weaknesses

The main issue that stood out to me was with regard to the implementation
details of the classification-based algorithms that were tested.
It would&rsquo;ve been great if pseudocode could&rsquo;ve been provided for the
classification versions of ReBRAC, IQL, and LB-SAC that the authors tested.
I perused the source code, and I am not convinced that the implementations are
truly proper adaptations of these algorithms: in particular, as actor-critic
algorithms, the actor networks are still trained by regression, and in some
cases, there is a sort of critic baseline that is trained by regression. For
instance, in the IQL adaptation, the $V$ function is still trained with
expectile regression (as in the original IQL)&#x2014;this function is then used to
update the $Q$ function via cross-entropy. So, altogether, I am not convinced
that the results observed in the experiments are fairly judging the influence of
classification losses for scaling RL. Similarly, since Farebrother et al.
considered only value-based methods, it is not clear to me that the discrepancy
in consistency/performance observed in this work is due to the offline nature of
the problem, or due to the adaptation of actor-critic algorithms.

Moreover, the work of Farebrother et al. never claims that classification is
&ldquo;plug-and-play&rdquo; (that I could find, at least). They claim that HL-Gauss is a
&ldquo;drop-in replacement&rdquo; for MSE, which it is (at least for value-based methods):
they simply replace the loss and change the network architecture, but I didn&rsquo;t
interpret their work to suggest that this works without hyperparameter tuning.

Furthermore, I believe the study could have been motivated better. In
particular, what is it about offline RL in particular that required a more
thorough examination of classification-based losses?

More specific details and minor issues are listed in the &ldquo;requested changes&rdquo;
section of the review.

---

> ### Author Response · Authors · 2024-10-05
>
> We thank reviewer for the detailed feedback on our work. We will upload the updated PDF after resolving all reviewers requests and notify about update after that.
>
> # Addressing weaknesses
> 1. We will provide  pseudo-code for our implementations in appendix.
> 2. "The actor networks are still trained by regression". We disagree that actors trained with regression objective. In all cases actors objectives are to maximize discounted returns predicted by critics which is different from regression task and a common approach for actor-critic algorithms. Putting classification objective here would not address any issues while for critics it  has potential to resolve some issues related to deep Q-learning which is a motivation of Farebrother et al.
> 3. "in the IQL adaptation, the function is still trained with expectile regression (as in the original IQL) — this function is then used to update the Q function via cross-entropy." Unfortunately, we were not able to come up with a good modification of expectile V-function loss with cross-entropy. It is not a trivial problem and from our point of view requires a separate research. Farebrother et al. were motivated on the problems related to the Q functions so we do the same. Note, that V function is updated through the Q function, so if cross-entropy helps Q function anyhow then it is expected to result into better V function too. Our result demonstrate that cross-entropy results in different Q functions that notably affect the performance (see AntMaze and Adroit results).
> 4. "it is not clear to me that the discrepancy in consistency/performance observed in this work is due to the offline nature of the problem, or due to the adaptation of actor-critic algorithms". The main motivation behind our work is to verify what are the consequences of applying cross-entropy for Q learning in offline RL actor-critic algorithms. We consider this aspect as a novelty aspect rather then the weakness.
> 5. "They claim that HL-Gauss is a “drop-in replacement” for MSE". Yes, but their study does not extensively comment on hyperparameters choice and only show the good choice for $\sigma$/$\zeta$ parameter (which we also find to work nicely) and claiming that bigger amount of bins should lead to better results which appeared to be inconsistent in our case. Our study demonstrates potential obstacles when applying cross-entropy.
> 6. "what is it about offline RL in particular that required a more thorough examination of classification-based losses?". Offline RL is special in this case due to the fact that most of the approaches modify the Q function, e.g. to be pessimistic which rises natural question on how classification should be applied in this cases.
>
> # Addressing changes
> 1. Will fix this, thanks
> 2. Will be fixed
> 3. We addressed this question in weaknesses discussion (2.). We are open for further discussion. But our study relies on findings from Farebrother et al which is about value functions and motivated by the specificity of offline RL approaches in this aspect (see Addressing weaknesses 6).
> 4. We are adding pseudo-codes in appendix. The influence of $\beta$ is a good point and actually it is why our study is conducted. We want  to understand whether cross-entropy can be applied when we have factors affecting the learned  Q functions. In case of ReBRAC it is not a problem as can be seen with it's results, however  it appeared to be a problem for LB-SAC.
> 5. See Addressing weaknesses 3. Pseudo-code will be added.
> 6. Pseudo-code will be added.
> 7. We added this for the explanation of the algorithm nature (how does it solves offline RL problems?) and it is taken from original study. It serves as a grounding for the algorithm usage as a representative example of the algorithm from Q-function regularized family.
> 8. It was referring to subsection 4.3. We are fixing that.
> 9. Here we meant the same grids that were used in ReBRAC work where IQL appeared to be the best competitor. We are  making this clearer in the updated script.
> 10. Those are presented in Tables 16, 17 and 18 in appendix. We didn't put them in the tables from the main text to keep tables readable.

---

> > ### Comment · Reviewer_DVpN · 2024-10-16
> >
> > Thanks to the authors for their reply and their amendments to the manuscript.
> >
> > I think I have a better appreciation for the contribution now. Still, I think the narrative of the paper is not quite right, but it is fixable. In particular, my personal takeaways from the paper are the following:
> >
> > 1. *Integrating classification losses into complex RL architectures requires care*. In particular, the issue is not that Farebrother et al. overclaimed --- it is that properly integrating classification losses into methods beyond FQI-style algorithms has not been figured out yet. Notably, I think the fact that e.g. the IQL implementation is not *truly* incorporating classification loss (due to the non-triviality of accomplishing this) should be emphasized (as the authors point out in their last response).
> > 2. *Offline RL is a more difficult setting to tackle with regard to classification losses*. Notably, with components such as regularization penalties that are crucial in offline RL, it is more difficult to pick proper parameters for modeling the action-values with categorical distributions.
> >
> > I strongly recommend for the authors to amend the document to reflect these takeaways, as they paint a much clearer picture about what the empirical results are actually proving.

---

### Author Response · Authors · 2024-10-05
**Paper update**

We updated the script according to the reviewers' requests.

---

### Author Response · Authors · 2024-11-10

We have published our camera-ready revision updated with a suggestions by adding "Limitations and Future Work" section which is hopefully resolves all of the remaining concerns. Please let us know if that is not the case.

---

### Decision · Action_Editor_jeEM · 2024-11-01

**Recommendation:** Accept with minor revision

**Comment:**

As detailed above, the paper received mixed reviews with regards to the audience criteria. Based on these comments, I suggest that the authors review the paper accordingly, incorporating the comments from all three reviewers.

**Audience:**

With regards to the audience claims, the reviews are mixed,  and I believe more work needs to be done to improve this aspect. One of the concerns is that the narrative of the paper may not be quite right.

To rectify it, one reviewer recommended to address the following: the algorithms do not completely replace regression with classification, so there is a need in an example of how this could be done and not just a 'naïve' adaptation of the methods.

Another comment from another reviewer suggested that the claim that the offline algorithms can be replaced with the classification loss may be less interesting than algorithm design choices, so the authors may need to present more insights into this aspect.

Finally, another concern is stated that the conclusion might not be insightful enough, so the authors are expected to reformulate it to guide future research. In particular, how would this paper help design new algorithms?

**Claims And Evidence:**

The reviewers suggest that the claims and evidence criteria are met.